# Screening Sinkhorn Algorithm for Regularized Optimal Transport

**Mokhtar Z. Alaya**
LITIS EA4108
University of Rouen Normandy
mokhtarzahdi.alaya@gmail.com

**Maxime Bérar**
LITIS EA4108
University of Rouen Normandy
maxime.berar@univ-rouen.fr

**Gilles Gasso**
LITIS EA4108
INSA, University of Rouen Normandy
gilles.gasso@insa-rouen.fr

**Alain Rakotomamonjy**
LITIS EA4108
University of Rouen Normandy
and Criteo AI Lab, Criteo Paris
alain.rakoto@insa-rouen.fr

## Abstract

We introduce in this paper a novel strategy for efficiently approximating the Sinkhorn distance between two discrete measures. After identifying neglectable components of the dual solution of the regularized Sinkhorn problem, we propose to screen those components by directly setting them at that value before entering the Sinkhorn problem. This allows us to solve a smaller Sinkhorn problem while ensuring approximation with provable guarantees. More formally, the approach is based on a new formulation of *dual of Sinkhorn divergence problem* and on the KKT optimality conditions of this problem, which enable identification of dual components to be screened. This new analysis leads to the SCREENKHORN algorithm. We illustrate the efficiency of SCREENKHORN on complex tasks such as dimensionality reduction and domain adaptation involving regularized optimal transport.

## 1 Introduction

Computing optimal transport (OT) distances between pairs of probability measures or histograms, such as the earth mover's distance [38, 33] and Monge-Kantorovich or Wasserstein distance [37], are currently generating an increasing attraction in different machine learning tasks [36, 27, 4, 21], statistics [17, 31, 14, 6, 16], and computer vision [8, 33, 35], among other applications [26, 32]. In many of these problems, OT exploits the geometric features of the objects at hand in the underlying spaces to be leveraged in comparing probability measures. This effectively leads to improved performance of methods that are oblivious to the geometry, for example the chi-squared distances or the Kullback-Leibler divergence. Unfortunately, this advantage comes at the price of an enormous computational cost of solving the OT problem, that can be prohibitive in large scale applications. For instance, the OT between two histograms with supports of equal size $n$ can be formulated as a linear programming problem that requires generally super $\mathcal{O}(n^{2.5})$ [28] arithmetic operations, which is problematic when $n$ becomes larger.

A remedy to the heavy computation burden of OT lies in a prevalent approach referred to as regularized OT [11] and operates by adding an entropic regularization penalty to the original problem. Such a regularization guarantees a unique solution, since the objective function is strongly convex, and a greater computational stability. More importantly, this regularized OT can be solved efficiently with celebrated matrix scaling algorithms, such as Sinkhorn's fixed point iteration method [34, 25, 22].

Several works have considered further improvements in the resolution of this regularized OT problem. A greedy version of Sinkhorn algorithm, called Greenkhorn [3], allows to select and update columns and rows that most violate the polytope constraints. Another approach based on low-rank approximation of the cost matrix using the Nyström method induces the Nys-Sink algorithm [2]. Other classical optimization algorithms have been considered for approximating the OT, for instance accelerated gradient descent [39, 13, 29], quasi-Newton methods [7, 12] and stochastic gradient descent [19, 1].

In this paper, we propose a novel technique for accelerating the Sinkhorn algorithm when computing regularized OT distance between discrete measures. Our idea is strongly related to a screening strategy when solving a *Lasso* problem in sparse supervised learning [20]. Based on the fact that a transport plan resulting from an OT problem is sparse or presents a large number of neglectable values [7], our objective is to identify the dual variables of an approximate Sinkhorn problem, that are smaller than a predefined threshold, and thus that can be safely removed before optimization while not altering too much the solution of the problem. Within this global context, our contributions are the following:

- From a methodological point of view, we propose a new formulation of the dual of the Sinkhorn divergence problem by imposing variables to be larger than a threshold. This formulation allows us to introduce sufficient conditions, computable beforehand, for a variable to strictly satisfy its constraint, leading then to a "screened" version of the dual of Sinkhorn divergence.
- We provide some theoretical analysis of the solution of the "screened" Sinkhorn divergence, showing that its objective value and the marginal constraint satisfaction are properly controlled as the number of screened variables decreases.
- From an algorithmic standpoint, we use a constrained L-BFGS-B algorithm [30, 9] but provide a careful analysis of the lower and upper bounds of the dual variables, resulting in a well-posed and efficient algorithm denoted as SCREENKHORN.
- Our empirical analysis depicts how the approach behaves in a simple Sinkhorn divergence computation context. When considered in complex machine learning pipelines, we show that SCREENKHORN can lead to strong gain in efficiency while not compromising on accuracy.

The remainder of the paper is organized as follow. In Section 2 we briefly review the basic setup of regularized discrete OT. Section 3 contains our main contribution, that is, the SCREENKHORN algorithm. Section 4 is devoted to theoretical guarantees for marginal violations of SCREENKHORN. In Section 5 we present numerical results for the proposed algorithm, compared with the state-of-art Sinkhorn algorithm as implemented in [15]. The proofs of theoretical results are postponed to the supplementary material as well as additional empirical results.

*Notation.* For any positive matrix $T \in \mathbb{R}^{n \times m}$, we define its entropy as $H(T) = -\sum_{i,j} T_{ij} \log(T_{ij})$. Let $r(T) = T\mathbf{1}_m \in \mathbb{R}^n$ and $c(T) = T^\top \mathbf{1}_n \in \mathbb{R}^m$ denote the rows and columns sums of $T$ respectively. The coordinates $r_i(T)$ and $c_j(T)$ denote the $i$-th row sum and the $j$-th column sum of $T$, respectively. The scalar product between two matrices denotes the usual inner product, that is $\langle T, W \rangle = \text{tr}(T^\top W) = \sum_{i,j} T_{ij} W_{ij}$, where $T^\top$ is the transpose of $T$. We write $\mathbf{1}$ (resp. $\mathbf{0}$) the vector having all coordinates equal to one (resp. zero). $\Delta(w)$ denotes the diag operator, such that if $w \in \mathbb{R}^n$, then $\Delta(w) = \text{diag}(w_1, \ldots, w_n) \in \mathbb{R}^{n \times n}$. For a set of indices $L = \{i_1, \ldots, i_k\} \subseteq \{1, \ldots, n\}$ satisfying $i_1 < \cdots < i_k$, we denote the complementary set of $L$ by $L^\complement = \{1, \ldots, n\} \backslash L$. We also denote $|L|$ the cardinality of $L$. Given a vector $w \in \mathbb{R}^n$, we denote $w_L = (w_{i_1}, \ldots, w_{i_k})^\top \in \mathbb{R}^k$ and its complementary $w_{L^\complement} \in \mathbb{R}^{n-k}$. The notation is similar for matrices; given another subset of indices $S = \{j_1, \ldots, j_l\} \subseteq \{1, \ldots, m\}$ with $j_1 < \cdots < j_l$, and a matrix $T \in \mathbb{R}^{n \times m}$, we use $T_{(L,S)}$, to denote the submatrix of $T$, namely the rows and columns of $T_{(L,S)}$ are indexed by $L$ and $S$ respectively. When applied to matrices and vectors, $\odot$ and $\oslash$ (Hadamard product and division) and exponential notations refer to elementwise operators. Given two real numbers $a$ and $b$, we write $a \vee b = \max(a, b)$ and $a \wedge b = \min(a, b)$.

## 2 Regularized discrete OT

We briefly expose in this section the setup of OT between two discrete measures. We then consider the case when those distributions are only available through a finite number of samples, that is $\mu = \sum_{i=1}^n \mu_i \delta_{x_i} \in \Sigma_n$ and $\nu = \sum_{j=1}^m \nu_i \delta_{y_j} \in \Sigma_m$, where $\Sigma_n$ is the probability simplex with $n$ bins,

namely the set of probability vectors in $\mathbb{R}_+^n$, i.e., $\Sigma_n = \{w \in \mathbb{R}_+^n : \sum_{i=1}^n w_i = 1\}$. We denote their probabilistic couplings set as $\Pi(\mu, \nu) = \{P \in \mathbb{R}_+^{n \times m}, P\mathbf{1}_m = \mu, P^\top \mathbf{1}_n = \nu\}$.

**Sinkhorn divergence.** Computing OT distance between the two discrete measures $\mu$ and $\nu$ amounts to solving a linear problem [24] given by

$$\mathcal{S}(\mu, \nu) = \min_{P \in \Pi(\mu, \nu)} \langle C, P \rangle,$$

where $P = (P_{ij}) \in \mathbb{R}^{n \times m}$ is called the transportation plan, namely each entry $P_{ij}$ represents the fraction of mass moving from $x_i$ to $y_j$, and $C = (C_{ij}) \in \mathbb{R}^{n \times m}$ is a cost matrix comprised of nonnegative elements and related to the energy needed to move a probability mass from $x_i$ to $y_j$. The entropic regularization of OT distances [11] relies on the addition of a penalty term as follows:

$$\mathcal{S}_\eta(\mu, \nu) = \min_{P \in \Pi(\mu, \nu)} \{\langle C, P \rangle - \eta H(P)\}, \tag{1}$$

where $\eta > 0$ is a regularization parameter. We refer to $\mathcal{S}_\eta(\mu, \nu)$ as the *Sinkhorn divergence* [11].

**Dual of Sinkhorn divergence.** Below we provide the derivation of the dual problem for the regularized OT problem (1). Towards this end, we begin with writing its Lagrangian dual function:

$$\mathscr{L}(P, w, z) = \langle C, P \rangle + \eta \langle \log P, P \rangle + \langle w, P\mathbf{1}_m - \mu \rangle + \langle z, P^\top \mathbf{1}_n - \nu \rangle.$$

The dual of Sinkhorn divergence can be derived by solving $\min_{P \in \mathbb{R}_+^{n \times m}} \mathscr{L}(P, w, z)$. It is easy to check that objective function $P \mapsto \mathscr{L}(P, w, z)$ is strongly convex and differentiable. Hence, one can solve the latter minimum by setting $\nabla_P \mathscr{L}(P, w, z)$ to $\mathbf{0}_{n \times m}$. Therefore, we get $P_{ij}^\star = \exp\left(-\frac{1}{\eta}(w_i + z_j + C_{ij}) - 1\right)$, for all $i = 1, \ldots, n$ and $j = 1, \ldots, m$. Plugging this solution, and setting the change of variables $u = -w/\eta - 1/2$ and $v = -z/\eta - 1/2$, the dual problem is given by

$$\min_{u \in \mathbb{R}^n, v \in \mathbb{R}^m} \left\{ \Psi(u, v) := \mathbf{1}_n^\top B(u, v) \mathbf{1}_m - \langle u, \mu \rangle - \langle v, \nu \rangle \right\}, \tag{2}$$

where $B(u, v) := \Delta(e^u) K \Delta(e^v)$ and $K := e^{-C/\eta}$ stands for the Gibbs kernel associated to the cost matrix $C$. We refer to problem (2) as the *dual of Sinkhorn divergence*. Then, the optimal solution $P^\star$ of the primal problem (1) takes the form $P^\star = \Delta(e^{u^\star}) K \Delta(e^{v^\star})$ where the couple $(u^\star, v^\star)$ satisfies:

$$(u^\star, v^\star) = \operatorname*{argmin}_{u \in \mathbb{R}^n, v \in \mathbb{R}^m} \{\Psi(u, v)\}.$$

Note that the matrices $\Delta(e^{u^\star})$ and $\Delta(e^{v^\star})$ are unique up to a constant factor [34]. Moreover, $P^\star$ can be solved efficiently by iterative Bregman projections [5] referred to as Sinkhorn iterations, and the method is referred to as SINKHORN algorithm which, recently, has been proven to achieve a near-$\mathcal{O}(n^2)$ complexity [3].

## 3 Screened dual of Sinkhorn divergence

**Motivation.** The key idea of our approach is motivated by the so-called *static screening test* [20] in supervised learning, which is a method able to safely identify inactive features, i.e., features that have zero components in the solution vector. Then, these inactive features can be removed from the optimization problem to reduce its scale. Before diving into detailed algorithmic analysis, let us present a brief illustration of how we adapt static screening test to the dual of Sinkhorn divergence. Towards this end, we define the convex set $\mathcal{C}_\alpha^r \subseteq \mathbb{R}^r$, for $r \in \mathbb{N}$ and $\alpha > 0$, by $\mathcal{C}_\alpha^r = \{w \in \mathbb{R}^r : e^{w_i} \geq \alpha\}$. In Figure 1, we plot $(e^{u^\star}, e^{v^\star})$ where $(u^\star, v^\star)$ is the pair solution of the dual of Sinkhorn divergence (2) in the particular case of:

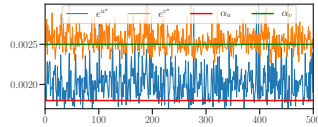

Figure 1: Plots of $(e^{u^\star}, e^{v^\star})$ with $(u^\star, v^\star)$ is the pair solution of dual of Sinkhorn divergence (2) and the thresholds $\alpha_u, \alpha_v$.

$n = m = 500, \eta = 1, \mu = \nu = \frac{1}{n}\mathbf{1}_n, x_i \sim \mathcal{N}((0,0)^\top, \left(\begin{smallmatrix} 1 & 0 \\ 0 & 1 \end{smallmatrix}\right)), y_j \sim \mathcal{N}((3,3)^\top, \left(\begin{smallmatrix} 1 & -0.8 \\ -0.8 & 1 \end{smallmatrix}\right))$ and the cost matrix $C$ corresponds to the pairwise euclidean distance, i.e., $C_{ij} = \|x_i - y_j\|_2$. We also plot two lines corresponding to $e^{u^\star} \equiv \alpha_u$ and $e^{v^\star} \equiv \alpha_v$ for some $\alpha_u > 0$ and $\alpha_v > 0$, choosing randomly and playing the role of thresholds to select indices to be discarded. If we are able to identify these indices before solving the problem, they can be fixed at the thresholds and removed then from the optimization procedure yielding an approximate solution.

**Static screening test.** Based on this idea, we define a so-called *approximate dual of Sinkhorn divergence*

$$\min_{u\in\mathcal{C}^n_{\frac{\varepsilon}{\kappa}},\, v\in\mathcal{C}^m_{\varepsilon\kappa}} \left\{ \Psi_\kappa(u,v) := \mathbf{1}_n^\top B(u,v)\mathbf{1}_m - \langle \kappa u, \mu\rangle - \langle \frac{v}{\kappa}, \nu\rangle \right\}, \tag{3}$$

which is simply a dual of Sinkhorn divergence with lower-bounded variables, where the bounds are $\alpha_u = \varepsilon\kappa^{-1}$ and $\alpha_v = \varepsilon\kappa$ with $\varepsilon > 0$ and $\kappa > 0$ being fixed numeric constants which values will be clear later. The new formulation (3) has the form of $(\kappa\mu, \nu/\kappa)$-scaling problem under constraints on the variables $u$ and $v$. Those constraints make the problem significantly different from the standard scaling-problems [23]. We further emphasize that $\kappa$ plays a key role in our screening strategy. Indeed, without $\kappa$, $e^u$ and $e^v$ can have inversely related scale that may lead in, for instance $e^u$ being too large and $e^v$ being too small, situation in which the screening test would apply only to coefficients of $e^u$ or $e^v$ and not for both of them. Moreover, it is clear that the approximate dual of Sinkhorn divergence coincides with the dual of Sinkhorn divergence (2) when $\varepsilon = 0$ and $\kappa = 1$. Intuitively, our hope is to gain efficiency in solving problem (3) compared to the original one in Equation (2) by avoiding optimization of variables smaller than the threshold and by identifying those that make the constraints active. More formally, the core of the static screening test aims at locating two subsets of indices $(I, J)$ in $\{1, \ldots, n\} \times \{1, \ldots, m\}$ satisfying: $e^{u_i} > \alpha_u$, and $e^{v_j} > \alpha_v$, for all $(i, j) \in I \times J$ and $e^{u_{i'}} = \alpha_u$, and $e^{v_{j'}} = \alpha_v$, for all $(i', j') \in I^{\complement} \times J^{\complement}$, namely $(u, v) \in \mathcal{C}^n_{\alpha_u} \times \mathcal{C}^m_{\alpha_v}$. The following key result states sufficient conditions for identifying variables in $I^{\complement}$ and $J^{\complement}$.

**Lemma 1.** *Let $(u^*, v^*)$ be an optimal solution of problem* (3)*. Define*

$$I_{\varepsilon,\kappa} = \left\{ i = 1, \ldots, n : \mu_i \geq \frac{\varepsilon^2}{\kappa} r_i(K) \right\}, J_{\varepsilon,\kappa} = \left\{ j = 1, \ldots, m : \nu_j \geq \kappa\varepsilon^2 c_j(K) \right\} \tag{4}$$

*Then one has $e^{u_i^*} = \varepsilon\kappa^{-1}$ and $e^{v_j^*} = \varepsilon\kappa$ for all $i \in I^{\complement}_{\varepsilon,\kappa}$ and $j \in J^{\complement}_{\varepsilon,\kappa}$.*

Proof of Lemma 1 is postponed to the supplementary material. It is worth to note that first order optimality conditions applied to $(u^*, v^*)$ ensure that if $e^{u_i^*} > \varepsilon\kappa^{-1}$ then $e^{u_i^*}(Ke^{v^*})_i = \kappa\mu_i$ and if $e^{v_j^*} > \varepsilon\kappa$ then $e^{v_j^*}(K^\top e^{u^*})_j = \kappa^{-1}\nu_j$, that correspond to the Sinkhorn marginal conditions [32] up to the scaling factor $\kappa$.

**Screening with a fixed number budget of points.** The approximate dual of Sinkhorn divergence is defined with respect to $\varepsilon$ and $\kappa$. As those parameters are difficult to interpret, we exhibit their relations with a fixed number budget of points from the supports of $\mu$ and $\nu$. In the sequel, we denote by $n_b \in \{1, \ldots, n\}$ and $m_b \in \{1, \ldots, m\}$ the number of points that are going to be optimized in problem (3), *i.e*, the points we cannot guarantee that $e^{u_i^*} = \varepsilon\kappa^{-1}$ and $e^{v_j^*} = \varepsilon\kappa$ .

Let us define $\xi \in \mathbb{R}^n$ and $\zeta \in \mathbb{R}^m$ to be the ordered decreasing vectors of $\mu \oslash r(K)$ and $\nu \oslash c(K)$ respectively, that is $\xi_1 \geq \xi_2 \geq \cdots \geq \xi_n$ and $\zeta_1 \geq \zeta_2 \geq \cdots \geq \zeta_m$. To keep only $n_b$-budget and $m_b$-budget of points, the parameters $\kappa$ and $\varepsilon$ satisfy $\varepsilon^2\kappa^{-1} = \xi_{n_b}$ and $\varepsilon^2\kappa = \zeta_{m_b}$. Hence

$$\varepsilon = (\xi_{n_b}\zeta_{m_b})^{1/4} \text{ and } \kappa = \sqrt{\frac{\zeta_{m_b}}{\xi_{n_b}}}. \tag{5}$$

This guarantees that $|I_{\varepsilon,\kappa}| = n_b$ and $|J_{\varepsilon,\kappa}| = m_b$ by construction. In addition, when $(n_b, m_b)$ tends to the full number budget of points $(n, m)$, the objective in problem (3) converges to the objective of dual of Sinkhorn divergence (2).

We are now in position to formulate the optimization problem related to the screened dual of Sinkhorn. Indeed, using the above analyses, any solution $(u^*, v^*)$ of problem (3) satisfies $e^{u_i^*} \geq \varepsilon\kappa^{-1}$ and $e^{v_j^*} \geq \varepsilon\kappa$ for all $(i, j) \in (I_{\varepsilon,\kappa} \times J_{\varepsilon,\kappa})$, and $e^{u_i^*} = \varepsilon\kappa^{-1}$ and $e^{v_j^*} = \varepsilon\kappa$ for all $(i, j) \in (I^{\complement}_{\varepsilon,\kappa} \times J^{\complement}_{\varepsilon,\kappa})$. Hence, we can restrict the problem (3) to variables in $I_{\varepsilon,\kappa}$ and $J_{\varepsilon,\kappa}$. This boils down to restricting the constraints feasibility $\mathcal{C}^n_{\frac{\varepsilon}{\kappa}} \cap \mathcal{C}^m_{\varepsilon\kappa}$ to the screened domain defined by $\mathcal{U}_{\text{sc}} \cap \mathcal{V}_{\text{sc}}$,

$$\mathcal{U}_{\text{sc}} = \{u \in \mathbb{R}^{n_b} : e^{u_{I_{\varepsilon,\kappa}}} \succeq \frac{\varepsilon}{\kappa}\mathbf{1}_{n_b}\} \text{ and } \mathcal{V}_{\text{sc}} = \{v \in \mathbb{R}^{m_b} : e^{v_{J_{\varepsilon,\kappa}}} \succeq \varepsilon\kappa\mathbf{1}_{m_b}\}$$

where the vector comparison $\succeq$ has to be understood elementwise. And, by replacing in Equation (3), the variables belonging to $(I^{\complement}_{\varepsilon,\kappa} \times J^{\complement}_{\varepsilon,\kappa})$ by $\varepsilon\kappa^{-1}$ and $\varepsilon\kappa$, we derive the *screened dual of Sinkhorn divergence problem* as

$$\min_{u\in\mathcal{U}_{\text{sc}},\, v\in\mathcal{V}_{\text{sc}}} \left\{ \Psi_{\varepsilon,\kappa}(u,v) \right\} \tag{6}$$

**Algorithm 1:** SCREENKHORN($C, \eta, \mu, \nu, n_b, m_b$)

---

**Step 1:** Screening pre-processing

1. $\xi \leftarrow \texttt{sort}(\mu \oslash r(K)), \zeta \leftarrow \texttt{sort}(\nu \oslash c(K));$ //(decreasing order)

2. $\varepsilon \leftarrow (\xi_{n_b}\zeta_{m_b})^{1/4}, \kappa \leftarrow \sqrt{\zeta_{m_b}/\xi_{n_b}};$

3. $I_{\varepsilon,\kappa} \leftarrow \{i = 1, \ldots, n : \mu_i \geq \varepsilon^2\kappa^{-1}r_i(K)\}, J_{\varepsilon,\kappa} \leftarrow \{j = 1, \ldots, m : \nu_j \geq \varepsilon^2\kappa c_j(K)\};$

4. $\underline{\mu} \leftarrow \min_{i \in I_{\varepsilon,\kappa}} \mu_i, \bar{\mu} \leftarrow \max_{i \in I_{\varepsilon,\kappa}} \mu_i, \underline{\nu} \leftarrow \min_{j \in J_{\varepsilon,\kappa}} \nu_i, \bar{\nu} \leftarrow \max_{j \in J_{\varepsilon,\kappa}} \nu_i;$

5. $\underline{u} \leftarrow \log\left(\frac{\varepsilon}{\kappa} \vee \frac{\underline{\mu}}{\varepsilon(m-m_b)+\varepsilon\vee\frac{\bar{\nu}}{n\varepsilon\kappa K_{\min}}m_b}\right), \bar{u} \leftarrow \log\left(\frac{\bar{\mu}}{m\varepsilon K_{\min}}\right);$

6. $\underline{v} \leftarrow \log\left(\varepsilon\kappa \vee \frac{\underline{\nu}}{\varepsilon(n-n_b)+\varepsilon\vee\frac{\kappa\bar{\mu}}{m\varepsilon K_{\min}}n_b}\right), \bar{v} \leftarrow \log\left(\frac{\bar{\nu}}{n\varepsilon K_{\min}}\right);$

7. $\bar{\theta} \leftarrow \texttt{stack}(\bar{u}\mathbf{1}_{n_b}, \bar{v}\mathbf{1}_{m_b}), \underline{\theta} \leftarrow \texttt{stack}(\underline{u}\mathbf{1}_{n_b}, \underline{v}\mathbf{1}_{m_b});$

**Step 2:** L-BFGS-B solver on the screened variables

8. $u^{(0)} \leftarrow \log(\varepsilon\kappa^{-1})\mathbf{1}_{n_b}, v^{(0)} \leftarrow \log(\varepsilon\kappa)\mathbf{1}_{m_b};$

9. $\hat{u}, \hat{v} \leftarrow \textsc{Restricted Sinkhorn}(u^{(0)}, v^{(0)}), \theta^{(0)} \leftarrow \texttt{stack}(\hat{u}, \hat{v});$

10. $\theta \leftarrow \text{L-BFGS-B}(\theta^{(0)}, \underline{\theta}, \bar{\theta});$

11. $\theta_u \leftarrow (\theta_1, \ldots, \theta_{n_b})^\top, \theta_v \leftarrow (\theta_{n_b+1}, \ldots, \theta_{n_b+m_b})^\top;$

12. $u_i^{sc} \leftarrow (\theta_u)_i$ if $i \in I_{\varepsilon,\kappa}$ and $u_i \leftarrow \log(\varepsilon\kappa^{-1})$ if $i \in I_{\varepsilon,\kappa}^{\complement};$

13. $v_j^{sc} \leftarrow (\theta_v)_j$ if $j \in J_{\varepsilon,\kappa}$ and $v_j \leftarrow \log(\varepsilon\kappa)$ if $j \in J_{\varepsilon,\kappa}^{\complement};$

14. **return** $B(u^{sc}, v^{sc})$.

---

where

$$\Psi_{\varepsilon,\kappa}(u,v) = (e^{u_{I_{\varepsilon,\kappa}}})^\top K_{(I_{\varepsilon,\kappa},J_{\varepsilon,\kappa})}e^{v_{J_{\varepsilon,\kappa}}} + \varepsilon\kappa(e^{u_{I_{\varepsilon,\kappa}}})^\top K_{(I_{\varepsilon,\kappa},J_{\varepsilon,\kappa}^{\complement})}\mathbf{1}_{m_b} + \varepsilon\kappa^{-1}\mathbf{1}_{n_b}^\top K_{(I_{\varepsilon,\kappa}^{\complement},J_{\varepsilon,\kappa})}e^{v_{J_{\varepsilon,\kappa}}}$$
$$- \kappa\mu_{I_{\varepsilon,\kappa}}^\top u_{I_{\varepsilon,\kappa}} - \kappa^{-1}\nu_{J_{\varepsilon,\kappa}}^\top v_{J_{\varepsilon,\kappa}} + \Xi$$

with $\Xi = \varepsilon^2 \sum_{i \in I_{\varepsilon,\kappa}^{\complement}, j \in J_{\varepsilon,\kappa}^{\complement}} K_{ij} - \kappa\log(\varepsilon\kappa^{-1})\sum_{i \in I_{\varepsilon,\kappa}^{\complement}} \mu_i - \kappa^{-1}\log(\varepsilon\kappa)\sum_{j \in J_{\varepsilon,\kappa}^{\complement}} \nu_j$.

The above problem uses only the restricted parts $K_{(I_{\varepsilon,\kappa},J_{\varepsilon,\kappa})}$, $K_{(I_{\varepsilon,\kappa},J_{\varepsilon,\kappa}^{\complement})}$, and $K_{(I_{\varepsilon,\kappa}^{\complement},J_{\varepsilon,\kappa})}$ of the Gibbs kernel $K$ for calculating the objective function $\Psi_{\varepsilon,\kappa}$. Hence, a gradient descent scheme will also need only those rows/columns of $K$. This is in contrast to Sinkhorn algorithm which performs alternating updates of all rows and columns of $K$. In summary, SCREENKHORN consists of two steps: the first one is a screening pre-processing providing the active sets $I_{\varepsilon,\kappa}, J_{\varepsilon,\kappa}$. The second one consists in solving Equation (6) using a constrained L-BFGS-B [9] for the stacked variable $\theta = (u_{I_{\varepsilon,\kappa}}, v_{J_{\varepsilon,\kappa}})$. Pseudocode of our proposed algorithm is shown in Algorithm 1. Note that in practice, we initialize the L-BFGS-B algorithm based on the output of a method, called RESTRICTED SINKHORN (see Algorithm 1 in the supplementary), which is a Sinkhorn-like algorithm applied to the active dual variables $\theta = (u_{I_{\varepsilon,\kappa}}, v_{J_{\varepsilon,\kappa}})$. While simple and efficient, the solution of this RESTRICTED SINKHORN algorithm does not satisfy the lower bound constraints of Problem (6) but provide a good candidate solution. Also note that L-BFGS-B handles box constraints on variables, but it becomes more efficient when these box bounds are carefully determined for problem (6). The following proposition (proof in supplementary material) expresses these bounds that are pre-calculated in the initialization step of SCREENKHORN.

**Proposition 1.** *Let $(u^{sc}, v^{sc})$ be an optimal pair solution of problem* (6) *and* $K_{\min} = \min\limits_{i \in I_{\varepsilon,\kappa}, j \in J_{\varepsilon,\kappa}} K_{ij}$.
*Then, one has*

$$\frac{\varepsilon}{\kappa} \vee \frac{\min_{i \in I_{\varepsilon,\kappa}} \mu_i}{\varepsilon(m - m_b) + \frac{\max_{j \in J_{\varepsilon,\kappa}} \nu_j}{n\varepsilon\kappa K_{\min}}m_b} \leq e^{u_i^{sc}} \leq \frac{\max_{i \in I_{\varepsilon,\kappa}} \mu_i}{m\varepsilon K_{\min}}, \qquad (7)$$

*and*

$$\varepsilon\kappa \vee \frac{\min_{j \in J_{\varepsilon,\kappa}} \nu_j}{\varepsilon(n - n_b) + \frac{\kappa\max_{i \in I_{\varepsilon,\kappa}} \mu_i}{m\varepsilon K_{\min}}n_b} \leq e^{v_j^{sc}} \leq \frac{\max_{j \in J_{\varepsilon,\kappa}} \nu_j}{n\varepsilon K_{\min}} \qquad (8)$$

*for all $i \in I_{\varepsilon,\kappa}$ and $j \in J_{\varepsilon,\kappa}$.*

# 4 Theoretical analysis and guarantees

This section is devoted to establishing theoretical guarantees for SCREENKHORN algorithm. We first define the screened marginals $\mu^{\text{sc}} = B(u^{\text{sc}}, v^{\text{sc}})\mathbf{1}_m$ and $\nu^{\text{sc}} = B(u^{\text{sc}}, v^{\text{sc}})^\top \mathbf{1}_n$. Our first theoretical result, Proposition 2, gives an upper bound of the screened marginal violations with respect to $\ell_1$-norm.

**Proposition 2.** *Let $(u^{sc}, v^{sc})$ be an optimal pair solution of problem* (6). *Then one has*

$$\|\mu - \mu^{sc}\|_1^2 = \mathcal{O}\Big(n_b c_\kappa + (n - n_b)\Big(\frac{\|C\|_\infty}{\eta} + \frac{m_b}{\sqrt{nm c_{\mu\nu}} K_{\min}^{3/2}} + \frac{m - m_b}{\sqrt{nm} K_{\min}} + \log\Big(\frac{\sqrt{nm}}{m_b c_{\mu\nu}^{5/2}}\Big)\Big)\Big) \quad (9)$$

*and*

$$\|\nu - \nu^{sc}\|_1^2 = \mathcal{O}\Big(m_b c_{\frac{1}{\kappa}} + (m - m_b)\Big(\frac{\|C\|_\infty}{\eta} + \frac{n_b}{\sqrt{nm c_{\mu\nu}} K_{\min}^{3/2}} + \frac{n - n_b}{\sqrt{nm} K_{\min}} + \log\Big(\frac{\sqrt{nm}}{n_b c_{\mu\nu}^{5/2}}\Big)\Big)\Big), \quad (10)$$

*where $c_z = z - \log z - 1$ for $z > 0$ and $c_{\mu\nu} = \underline{\mu} \wedge \underline{\nu}$ with $\underline{\mu} = \min_{i \in I_{\varepsilon,\kappa}} \mu_i$ and $\underline{\nu} = \min_{j \in J_{\varepsilon,\kappa}} \nu_j$.*

Proof of Proposition 2 is presented in supplementary material and it is based on first order optimality conditions for problem (6) and on a generalization of Pinsker inequality (see Lemma 1 in supplementary).

Our second theoretical result, Proposition 3, is an upper bound of the difference between objective values of SCREENKHORN and dual of Sinkhorn divergence (2).

**Proposition 3.** *Let $(u^{sc}, v^{sc})$ be an optimal pair solution of problem* (6) *and $(u^\star, v^\star)$ is the pair solution of dual of Sinkhorn divergence* (2). *Then we have*

$$\Psi_{\varepsilon,\kappa}(u^{sc}, v^{sc}) - \Psi(u^\star, v^\star) = \mathcal{O}\big(R(\|\mu - \mu^{sc}\|_1 + \|\nu - \nu^{sc}\|_1 + \omega_\kappa)\big).$$

*where $R = \frac{\|C\|_\infty}{\eta} + \log\big(\frac{(n \vee m)^2}{nm c_{\mu\nu}^{7/2}}\big)$ and $\omega_\kappa = |1 - \kappa|\|\mu^{sc}\|_1 + |1 - \kappa^{-1}|\|\nu^{sc}\|_1 + |1 - \kappa| + |1 - \kappa^{-1}|.$*

Proof of Proposition 3 is exposed in the supplementary material. Comparing to some other analysis results of this quantity, see for instance Lemma 2 in [13] and Lemma 3.1 in [29], our bound involves an additional term $\omega_\kappa$ (with $\omega_1 = 0$). To better characterize $\omega_\kappa$, a control of the $\ell_1$-norms of the screened marginals $\mu^{\text{sc}}$ and $\nu^{\text{sc}}$ are given in Lemma 2 in the supplementary material.

# 5 Numerical experiments

In this section, we present some numerical analyses of our SCREENKHORN algorithm and show how it behaves when integrated into some complex machine learning pipelines.

## 5.1 Setup

We have implemented our SCREENKHORN algorithm in Python and used the L-BFGS-B of Scipy. Regarding the machine-learning based comparison, we have based our code on the ones of Python Optimal Transport toolbox (POT) [15] and just replaced the `sinkhorn` function call with a `screenkhorn` one. We have considered the POT's default SINKHORN stopping criterion parameters and for SCREENKHORN, the L-BFGS-B algorithm is stopped when the largest component of the projected gradient is smaller than $10^{-6}$, when the number of iterations or the number of objective function evaluations reach $10^5$. For all applications, we have set $\eta = 1$ unless otherwise specified.

## 5.2 Analysing on toy problem

We compare SCREENKHORN to SINKHORN as implemented in POT toolbox[1] on a synthetic example. The dataset we use consists of source samples generated from a bi-dimensional gaussian mixture and target samples following the same distribution but with different gaussian means. We consider an unsupervised domain adaptation using optimal transport with entropic regularization. Several settings are explored: different values of $\eta$, the regularization parameter, the allowed budget $\frac{n_b}{n} = \frac{m_b}{m}$

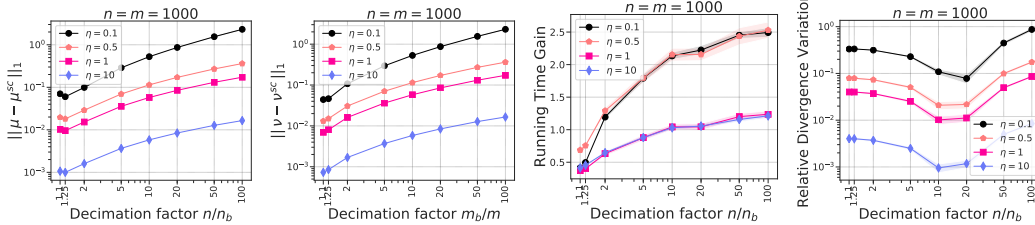

Figure 2: Empirical evaluation of SCREENKHORN vs SINKHORN for normalized cost matrix *i.e.* $\|C\|_\infty = 1$. (most-lefts): marginal violations in relation with the budget of points on $n$ and $m$. (center-right) ratio of computation times $\frac{T_{\text{SINKHORN}}}{T_{\text{SCREENKHORN}}}$ and, (right) relative divergence variation. The results are averaged over 30 trials.

ranging from $0.01$ to $0.99$, different values of $n$ and $m$. We empirically measure marginal violations as the norms $\|\mu - \mu^{\text{sc}}\|_1$ and $\|\nu - \nu^{\text{sc}}\|_1$, running time expressed as $\frac{T_{\text{SINKHORN}}}{T_{\text{SCREENKHORN}}}$ and the relative divergence difference $|\langle C, P^\star \rangle - \langle C, P^{\text{sc}} \rangle| / \langle C, P^\star \rangle$ between SCREENKHORN and SINKHORN, where $P^\star = \Delta(e^{u^\star}) K \Delta(e^{v^\star})$ and $P^{\text{sc}} = \Delta(e^{u^{\text{sc}}}) K \Delta(e^{v^{\text{sc}}})$. Figure 2 summarizes the observed behaviors of both algorithms under these settings. We choose to only report results for $n = m = 1000$ as we get similar findings for other values of $n$ and $m$.

SCREENKHORN provides good approximation of the marginals $\mu$ and $\nu$ for "high" values of the regularization parameter $\eta$ ($\eta > 1$). The approximation quality diminishes for small $\eta$. As expected $\|\mu - \mu^{\text{sc}}\|_1$ and $\|\nu - \nu^{\text{sc}}\|_1$ converge towards zero when increasing the budget of points. Remarkably marginal violations are almost negligible whatever the budget for high $\eta$. According to computation gain, SCREENKHORN is almost 2 times faster than SINKHORN at high decimation factor $n/n_b$ (low budget) while the reverse holds when $n/n_b$ gets close to 1. Computational benefit of SCREENKHORN also depends on $\eta$ with appropriate values $\eta \leq 1$. Finally except for $\eta = 0.1$ SCREENKHORN achieves a divergence $\langle C, P \rangle$ close to the one of Sinkhorn showing that our static screening test provides a reasonable approximation of the Sinkhorn divergence. As such, we believe that SCREENKHORN will be practically useful in cases where modest accuracy on the divergence is sufficient. This may be the case of a loss function for a gradient descent method (see next section).

### 5.3 Integrating SCREENKHORN into machine learning pipelines

Here, we analyse the impact of using SCREENKHORN instead of SINKHORN in a complex machine learning pipeline. Our two applications are a dimensionality reduction technique, denoted as Wasserstein Discriminant Analysis (WDA), based on Wasserstein distance approximated through Sinkhorn divergence [16] and a domain-adaptation using optimal transport mapping [10], named OTDA.

WDA aims at finding a linear projection which minimize the ratio of distance between intra-class samples and distance inter-class samples, where the distance is understood in a Sinkhorn divergence sense. We have used a toy problem involving Gaussian classes with 2 discriminative features and 8 noisy features and the MNIST dataset. For the former problem, we aim at find the best two-dimensional linear subspace in a WDA sense whereas for MNIST, we look for a subspace of dimension 20 starting from the original 728 dimensions. Quality of the retrieved subspace are evaluated using classification task based on a 1-nearest neighbour approach.

Figure 3 presents the average gain (over 30 trials) in computational time we get as the number of examples evolve and for different decimation factors of the SCREENKHORN problem. Analysis of the quality of the subspace have been deported to the supplementary material (see Figure 2), but we can remark a small loss of performance of SCREENKHORN for the toy problem, while for MNIST, accuracies are equivalent regardless of the decimation factor. We can note that the minimal gains are respectively 2 and 4.5 for the toy and MNIST problem whereas the maximal gain for 4000 samples is slightly larger than an order of magnitude.

For the OT based domain adaptation problem, we have considered the OTDA with $\ell_{\frac{1}{2},1}$ group-lasso regularizer that helps in exploiting available labels in the source domain. The problem is solved using a majorization-minimization approach for handling the non-convexity of the problem. Hence, at each

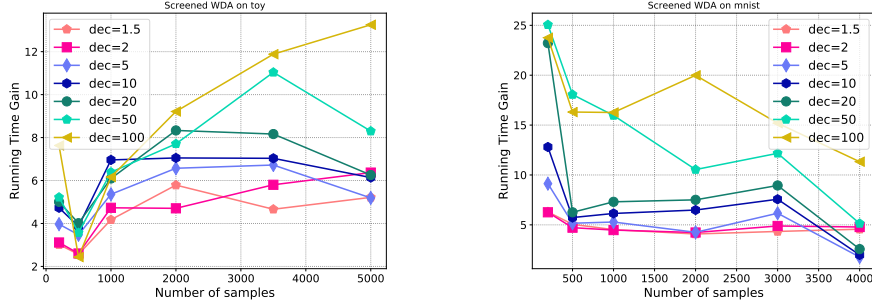

Figure 3: Wasserstein Discriminant Analysis : running time gain for (left) a toy dataset and (right) MNIST as a function of the number of examples and the data decimation factor in SCREENKHORN.

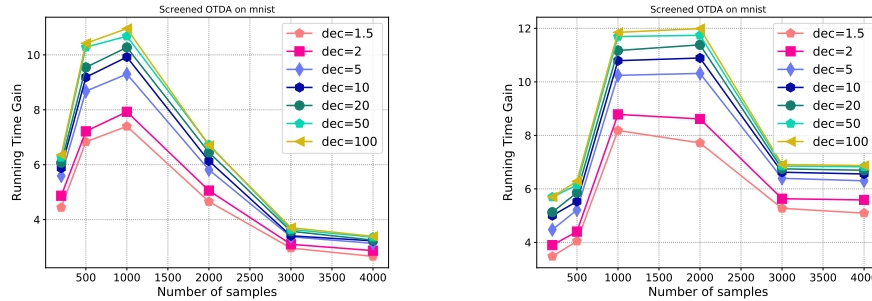

Figure 4: OT Domain adaptation : running time gain for MNIST as a function of the number of examples and the data decimation factor in SCREENKHORN. Group-lasso hyperparameter values (left) 1. (right) 10.

iteration, a SINKHORN/SCREENKHORN has to be computed and the number of iteration is sensitive to the regularizer strength. As a domain-adaptation problem, we have used a MNIST to USPS problem in which features have been computed from the first layers of a domain adversarial neural networks [18] before full convergence of the networks (so as to leave room for OT adaptation). Figure 4 reports the gain in running time for 2 different values of the group-lasso regularizer hyperparameter, while the curves of performances are reported in the supplementary material. We can note that for all the SCREENKHORN with different decimation factors, the gain in computation goes from a factor of 4 to 12, without any loss of the accuracy performance.

## 6 Conclusion

The paper introduces a novel efficient approximation of the Sinkhorn divergence based on a screening strategy. Screening some of the Sinkhorn dual variables has been made possible by defining a novel constrained dual problem and by carefully analyzing its optimality conditions. From the latter, we derived some sufficient conditions depending on the ground cost matrix, that some dual variables are smaller than a given threshold. Hence, we need just to solve a restricted dual Sinkhorn problem using an off-the-shelf L-BFGS-B algorithm. We also provide some theoretical guarantees of the quality of the approximation with respect to the number of variables that have been screened. Numerical experiments show the behaviour of our SCREENKHORN algorithm and computational time gain it can achieve when integrated in some complex machine learning pipelines.

### Acknowledgments

This work was supported by grants from the Normandie Projet GRR-DAISI, European funding FEDER DAISI and OATMIL ANR-17-CE23-0012 Project of the French National Research Agency (ANR).

## Footnotes

[1] `https://pot.readthedocs.io/en/stable/index.html`

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
