[Supplementary Material]

# Supplementary material for "Screening Sinkhorn Algorithm for Regularized Optimal Transport"

**Mokhtar Z. Alaya**
LITIS EA4108
University of Rouen
mokhtarzahdi.alaya@gmail.com

**Maxime Bérar**
LITIS EA4108
University of Rouen
maxime.berar@univ-rouen.fr

**Gilles Gasso**
LITIS EA4108
INSA, University of Rouen
gilles.gasso@insa-rouen.fr

**Alain Rakotomamonjy**
LITIS EA4108
University of Rouen
and Criteo AI Labs, Criteo Paris
alain.rakoto@insa-rouen.fr

## 1 Proof of Lemma 1

Since the objective function $\Psi_\kappa$ is convex with respect to $(u, v)$, the set of optima of problem (3) is non empty. Introducing two dual variables $\lambda \in \mathbb{R}_+^n$ and $\beta \in \mathbb{R}_+^m$ for each constraint, the Lagrangian of problem (3) reads as

$$\mathscr{L}(u, v, \lambda, \beta) = \frac{\varepsilon}{\kappa}\langle\lambda, \mathbf{1}_n\rangle + \varepsilon\kappa\langle\beta, \mathbf{1}_m\rangle + \mathbf{1}_n^\top B(u,v)\mathbf{1}_m - \langle\kappa u, \mu\rangle - \langle\frac{v}{\kappa}, \nu\rangle - \langle\lambda, e^u\rangle - \langle\beta, e^v\rangle$$

First order conditions then yield that the Lagrangian multiplicators solutions $\lambda^*$ and $\beta^*$ satisfy

$$\nabla_u \mathscr{L}(u^*, v^*, \lambda^*, \beta^*) = e^{u^*} \odot (Ke^{v^*} - \lambda^*) - \kappa\mu = \mathbf{0}_n,$$

$$\text{and } \nabla_v \mathscr{L}(u^*, v^*, \lambda^*, \beta^*) = e^{v^*} \odot (K^\top e^{u^*} - \beta) - \frac{\nu}{\kappa} = \mathbf{0}_m$$

which leads to

$$\lambda^* = Ke^{v^*} - \kappa\mu \oslash e^{u^*} \text{ and } \beta^* = K^\top e^{u^*} - \nu \oslash \kappa e^{v^*}$$

For all $i = 1, \ldots, n$ we have that $e^{u_i^*} \geq \frac{\varepsilon}{\kappa}$. Further, the condition on the dual variable $\lambda_i^* > 0$ ensures that $e^{u_i^*} = \frac{\varepsilon}{\kappa}$ and hence $i \in I_{\varepsilon,\kappa}^{\complement}$. We have that $\lambda_i^* > 0$ is equivalent to $e^{u_i^*} r_i(K) e^{v_j^*} > \kappa\mu_i$ which is satisfied when $\varepsilon^2 r_i(K) > \kappa\mu_i$. In a symmetric way we can prove the same statement for $e^{v_j^*}$.

## 2 Proof of Proposition 1

We prove only the first statement (7) and similarly we can prove the second one (8). For all $i \in I_{\varepsilon,\kappa}$, we have $e^{u_i^{\text{sc}}} > \frac{\varepsilon}{\kappa}$ or $e^{u_i^{\text{sc}}} = \frac{\varepsilon}{\kappa}$. In one hand, if $e^{u_i^{\text{sc}}} > \frac{\varepsilon}{\kappa}$ then according to the optimality conditions $\lambda_i^{\text{sc}} = 0$, which implies $e^{u_i^{\text{sc}}} \sum_{j=1}^m K_{ij} e^{v_j^{\text{sc}}} = \kappa\mu_i$. In another hand, we have

$$e^{u_i^{\text{sc}}} \min_{i,j} K_{ij} \sum_{j=1}^m e^{v_j^{\text{sc}}} \leq e^{u_i^{\text{sc}}} \sum_{j=1}^m K_{ij} e^{v_j^{\text{sc}}} = \kappa\mu_i.$$

We further observe that $\sum_{j=1}^m e^{v_j^{\text{sc}}} = \sum_{j \in J_{\varepsilon,\kappa}} e^{v_j^{\text{sc}}} + \sum_{j \in J_{\varepsilon,\kappa}^{\complement}} e^{v_j^{\text{sc}}} \geq \varepsilon\kappa|J_{\varepsilon,\kappa}| + \varepsilon\kappa|J_{\varepsilon,\kappa}^{\complement}| = \varepsilon\kappa m$. Then

$$\max_{i \in I_{\varepsilon,\kappa}} e^{u_i^{\text{sc}}} \leq \frac{\max_{i \in I_{\varepsilon,\kappa}} \mu_i}{m\varepsilon K_{\min}}.$$

Analogously, one can obtain for all $j \in J_{\varepsilon, \kappa}$

$$\max_{j \in J_{\varepsilon,\kappa}} e^{v_j^{\mathrm{sc}}} \leq \frac{\max_{j \in J_{\varepsilon,\kappa}} \nu_j}{n \varepsilon K_{\min}}. \tag{1}$$

Now, since $K_{ij} \leq 1$, we have

$$e^{u_i^{\mathrm{sc}}} \sum_{j=1}^{m} e^{v_j^{\mathrm{sc}}} \geq e^{u_i^{\mathrm{sc}}} \sum_{j=1}^{m} K_{ij} e^{v_j^{\mathrm{sc}}} = \kappa \mu_i.$$

Using (1), we get

$$\sum_{j=1}^{m} e^{v_j^{\mathrm{sc}}} = \sum_{j \in J_{\varepsilon,\kappa}} e^{v_j^{\mathrm{sc}}} + \sum_{j \in J_{\varepsilon,\kappa}^{\complement}} e^{v_j^{\mathrm{sc}}} \leq \varepsilon \kappa |J_{\varepsilon,\kappa}^{\complement}| + \frac{\max_{j \in J_{\varepsilon,\kappa}} \nu_j}{n \varepsilon K_{\min}} |J_{\varepsilon,\kappa}|.$$

Therefore,

$$\min_{i \in I_{\varepsilon,\kappa}} e^{u_i^{\mathrm{sc}}} \geq \frac{\varepsilon}{\kappa} \vee \frac{\kappa \min_{I_{\varepsilon,\kappa}} \mu_i}{\varepsilon \kappa (m - m_b) + \frac{\max_{j \in J_{\varepsilon,\kappa}} \nu_j}{n \varepsilon K_{\min}} m_b}.$$

## 3 Proof of Proposition 2

We define the distance function $\varrho : \mathbb{R}_+ \times \mathbb{R}_+ \mapsto [0, \infty]$ by $\varrho(a, b) = b - a + a \log(\frac{a}{b})$. While $\varrho$ is not a metric, it is easy to see that $\varrho$ is not nonnegative and satisfies $\varrho(a, b) = 0$ iff $a = b$. The violations are computed through the following function:

$$d_{\varrho}(\gamma, \beta) = \sum_{i=1}^{n} \varrho(\gamma_i, \beta_i), \text{ for } \gamma, \beta \in \mathbb{R}_+^n.$$

Note that if $\gamma, \beta$ are two vectors of positive entries, $d_{\varrho}(\gamma, \beta)$ will return some measurement on how far they are from each other. The next Lemma is from [1] (see Lemma 7 herein).

**Lemma 1.** *For any $\gamma, \beta \in \mathbb{R}_+^n$, the following generalized Pinsker inequality holds*

$$\|\gamma - \beta\|_1 \leq \sqrt{7(\|\gamma\|_1 \wedge \|\beta\|_1) d_{\varrho}(\gamma, \beta)}.$$

The optimality conditions for $(u^{\mathrm{sc}}, v^{\mathrm{sc}})$ entails

$$\mu_i^{\mathrm{sc}} = \begin{cases} e^{u_i^{\mathrm{sc}}} \sum_{j=1}^{m} K_{ij} e^{v_j^{\mathrm{sc}}}, & \text{if } i \in I_{\varepsilon,\kappa}, \\ \frac{\varepsilon}{\kappa} \sum_{j=1}^{m} K_{ij} e^{v_j^{\mathrm{sc}}}, & \text{if } i \in I_{\varepsilon,\kappa}^{\complement} \end{cases} = \begin{cases} \kappa \mu_i, & \text{if } i \in I_{\varepsilon,\kappa}, \\ \frac{\varepsilon}{\kappa} \sum_{j=1}^{m} K_{ij} e^{v_j^{\mathrm{sc}}}, & \text{if } i \in I_{\varepsilon,\kappa}^{\complement}, \end{cases} \tag{2}$$

and

$$\nu_j^{\mathrm{sc}} = \begin{cases} e^{v_j^{\mathrm{sc}}} \sum_{i=1}^{n} K_{ij} e^{u_i^{\mathrm{sc}}}, & \text{if } j \in J_{\varepsilon,\kappa}, \\ \varepsilon \kappa \sum_{i=1}^{n} K_{ij} e^{u_i^{\mathrm{sc}}}, & \text{if } j \in J_{\varepsilon,\kappa}^{\complement} \end{cases} = \begin{cases} \frac{\nu_j}{\kappa}, & \text{if } j \in J_{\varepsilon,\kappa}, \\ \varepsilon \kappa \sum_{i=1}^{n} K_{ij} e^{u_i^{\mathrm{sc}}}, & \text{if } j \in J_{\varepsilon,\kappa}^{\complement}. \end{cases} \tag{3}$$

By (2), we have

$$d_{\varrho}(\mu, \mu^{\mathrm{sc}}) = \sum_{i=1}^{n} \mu_i^{\mathrm{sc}} - \mu_i + \mu_i \log\left(\frac{\mu_i}{\mu_i^{\mathrm{sc}}}\right)$$

$$= \sum_{i \in I_{\varepsilon,\kappa}} (\kappa - 1)\mu_i - \mu_i \log(\kappa) + \sum_{i \in I_{\varepsilon,\kappa}^{\complement}} \frac{\varepsilon}{\kappa} \sum_{j=1}^{m} K_{ij} e^{v_j^{\mathrm{sc}}} - \mu_i + \mu_i \log\left(\frac{\mu_i}{\frac{\varepsilon}{\kappa} \sum_{j=1}^{m} K_{ij} e^{v_j^{\mathrm{sc}}}}\right)$$

$$= \sum_{i \in I_{\varepsilon,\kappa}} (\kappa - \log(\kappa) - 1)\mu_i + \sum_{i \in I_{\varepsilon,\kappa}^{\complement}} \frac{\varepsilon}{\kappa} \sum_{j=1}^{m} K_{ij} e^{v_j^{\mathrm{sc}}} - \mu_i + \mu_i \log\left(\frac{\mu_i}{\frac{\varepsilon}{\kappa} \sum_{j=1}^{m} K_{ij} e^{v_j^{\mathrm{sc}}}}\right).$$

Now by (8), we have in one hand

$$\sum_{i \in I_{\varepsilon,\kappa}^{\complement}} \frac{\varepsilon}{\kappa} \sum_{j=1}^{m} K_{ij} e^{v_j^{\mathrm{sc}}} = \sum_{i \in I_{\varepsilon,\kappa}^{\complement}} \frac{\varepsilon}{\kappa} \Big( \sum_{j \in J_{\varepsilon,\kappa}} K_{ij} e^{v_j^{\mathrm{sc}}} + \varepsilon\kappa \sum_{j \in J_{\varepsilon,\kappa}^{\complement}} K_{ij} \Big)$$

$$\leq \sum_{i \in I_{\varepsilon,\kappa}^{\complement}} \frac{\varepsilon}{\kappa} \Big( m_b \max_{i,j} K_{ij} \frac{\max_{j \in J_{\varepsilon,\kappa}} \nu_j}{n\varepsilon K_{\min}} + (m - m_b)\varepsilon\kappa \max_{i,j} K_{ij} \Big)$$

$$\leq (n - n_b) \Big( \frac{m_b \max_j \nu_j}{n\kappa K_{\min}} + (m - m_b)\varepsilon^2 \Big).$$

On the other hand, we get

$$\frac{\varepsilon}{\kappa} \sum_{j=1}^{m} K_{ij} e^{v_j^{\mathrm{sc}}} = \frac{\varepsilon}{\kappa} \Big( \sum_{j \in J_{\varepsilon,\kappa}} K_{ij} e^{v_j^{\mathrm{sc}}} + \varepsilon\kappa \sum_{j \in J_{\varepsilon,\kappa}^{\complement}} K_{ij} \Big)$$

$$\geq m_b K_{\min} \frac{m\varepsilon^2 K_{\min} \min_{j \in J_{\varepsilon,\kappa}} \nu_j}{\kappa((n - n_b)m\varepsilon^2 K_{\min} + m\varepsilon^2 K_{\min} + n_b\kappa \max_{i \in I_{\varepsilon,\kappa}} \mu_i)}$$

$$+ \varepsilon^2(m - m_b)K_{\min}$$

$$\geq \frac{mm_b\varepsilon^2(K_{\min})^2 \min_{j \in J_{\varepsilon,\kappa}} \nu_j}{\kappa((n - n_b)m\varepsilon^2 K_{\min} + m\varepsilon^2 K_{\min} + n_b\kappa \max_{i \in I_{\varepsilon,\kappa}} \mu_i)}$$

$$+ \varepsilon^2(m - m_b)K_{\min}$$

$$\geq \frac{mm_b\varepsilon^2 K_{\min}^2 \min_{j \in J_{\varepsilon,\kappa}} \nu_j}{\kappa((n - n_b)m\varepsilon^2 K_{\min} + m\varepsilon^2 K_{\min} + n_b\kappa \max_{i \in I_{\varepsilon,\kappa}} \mu_i)}.$$

Then

$$\frac{1}{\frac{\varepsilon}{\kappa} \sum_{j=1}^{m} K_{ij} e^{v_j^{\mathrm{sc}}}} \leq \frac{\kappa((n - n_b)m\varepsilon^2 K_{\min} + m\varepsilon^2 K_{\min} + n_b\kappa \max_{i \in I_{\varepsilon,\kappa}} \mu_i)}{mm_b\varepsilon^2 K_{\min}^2 \min_{j \in J_{\varepsilon,\kappa}} \nu_j}$$

$$\leq \frac{\kappa(n - n_b + 1)}{m_b K_{\min} \min_{j \in J_{\varepsilon,\kappa}} \nu_j} + \frac{n_b\kappa^2 \max_{i \in I_{\varepsilon,\kappa}} \mu_i}{mm_b\varepsilon^2 K_{\min}^2 \min_{j \in J_{\varepsilon,\kappa}} \nu_j}.$$

It entails

$$\sum_{i \in I_{\varepsilon,\kappa}^{\complement}} \frac{\varepsilon}{\kappa} \sum_{j=1}^{m} K_{ij} e^{v_j^{\mathrm{sc}}} - \mu_i + \mu_i \log \Big( \frac{\mu_i}{\frac{\varepsilon}{\kappa} \sum_{j=1}^{m} K_{ij} e^{v_j^{\mathrm{sc}}}} \Big)$$

$$\leq (n - n_b) \Big( \frac{m_b}{n\kappa K_{\min}} + (m - m_b)\varepsilon^2 - \min_i \mu_i$$

$$+ \max_i \mu_i \log \Big( \frac{\kappa(n - n_b + 1) \max_i \mu_i}{m_b K_{\min} \min_{j \in J_{\varepsilon,\kappa}} \nu_j} + \frac{n_b\kappa^2(\max_i \mu_i)^2}{mm_b\varepsilon^2 K_{\min}^2 \min_{j \in J_{\varepsilon,\kappa}} \nu_j} \Big) \Big).$$

Therefore

$$d_{\varrho}(\mu, \mu^{\mathrm{sc}}) \leq n_b c_\kappa \max_i \mu_i + (n - n_b) \Big( \frac{m_b \max_j \nu_j}{n\kappa K_{\min}} + (m - m_b)\varepsilon^2 - \min_i \mu_i$$

$$+ \max_i \mu_i \log \Big( \frac{\kappa(n - n_b + 1) \max_i \mu_i}{m_b K_{\min} \min_{j \in J_{\varepsilon,\kappa}} \nu_j} + \frac{n_b\kappa^2(\max_i \mu_i)^2}{mm_b\varepsilon^2 K_{\min}^2 \min_{j \in J_{\varepsilon,\kappa}} \nu_j} \Big).$$

Finally, by Lemma 1 we obtain

$$\|\mu - \mu^{\mathrm{sc}}\|_1^2 \leq n_b c_\kappa \max_i \mu_i + 7(n - n_b) \Big( \frac{m_b \max_j \nu_j}{n\kappa K_{\min}} + (m - m_b)\varepsilon^2 - \min_i \mu_i$$

$$+ \max_i \mu_i \log \Big( \frac{\kappa(n - n_b + 1) \max_i \mu_i}{m_b K_{\min} \min_{j \in J_{\varepsilon,\kappa}} \nu_j} + \frac{n_b\kappa^2(\max_i \mu_i)^2}{mm_b\varepsilon^2 K_{\min}^2 \min_{j \in J_{\varepsilon,\kappa}} \nu_j} \Big).$$

Following the same lines as above, we also have

$$\|\nu - \nu^{\mathrm{sc}}\|_1^2 \leq m_b c_{\frac{1}{\kappa}} \max_i \mu_i + 7(m - m_b)\left(\frac{n_b \kappa \max_i \mu_i}{m K_{\min}} + (n - n_b)\varepsilon^2 - \min_j \nu_j\right.$$
$$+ \max_j \nu_j \log\left(\frac{(m - m_b + 1)\max_j \nu_j}{n_b \kappa K_{\min} \min_{i \in I_{\varepsilon,\kappa}} \mu_i} + \frac{m_b(\max_j \nu_j)^2}{n n_b \varepsilon^2 \kappa^2 K_{\min}^2 \min_{i \in I_{\varepsilon,\kappa}} \mu_i}\right).$$

To get the closed forms (9) and (10), we used the following facts:

**Remark 1.** *We have* $\log(1/K_{\min}^r) = r\|C\|_\infty/\eta$, *for every* $r \in \mathbb{N}$. *Using* (5), *we further derive:* $\varepsilon = \mathcal{O}((mnK_{\min}^2)^{-1/4})$, $\kappa = \mathcal{O}(\sqrt{m/(nc_{\mu\nu}K_{\min})})$, $\kappa^{-1} = \mathcal{O}(\sqrt{n/(mK_{\min}c_{\mu\nu})})$, $(\kappa/\varepsilon)^2 = \mathcal{O}(m^{3/2}/\sqrt{nK_{\min}}(c_{\mu\nu})^{3/2})$, *and* $(\varepsilon\kappa)^{-2} = \mathcal{O}(n^{3/2}/\sqrt{mK_{\min}}c_{\mu\nu}^{3/2})$.

## 4 Proof of Proposition 3

We first define $\widetilde{K}$ a rearrangement of $K$ with respect to the active sets $I_{\varepsilon,\kappa}$ and $J_{\varepsilon,\kappa}$ as follows:

$$\widetilde{K} = \begin{bmatrix} K_{(I_{\varepsilon,\kappa}, J_{\varepsilon,\kappa})} & K_{(I_{\varepsilon,\kappa}, J_{\varepsilon,\kappa}^{\complement})} \\ K_{(I_{\varepsilon,\kappa}^{\complement}, J_{\varepsilon,\kappa})} & K_{(I_{\varepsilon,\kappa}^{\complement}, J_{\varepsilon,\kappa}^{\complement})} \end{bmatrix}.$$

Setting $\dot{\mu} = (\mu_{I_{\varepsilon,\kappa}}^\top, \mu_{I_{\varepsilon,\kappa}^{\complement}}^\top)^\top$, $\dot{\nu} = (\nu_{J_{\varepsilon,\kappa}}^\top, \nu_{J_{\varepsilon,\kappa}^{\complement}}^\top)^\top$ and for each vectors $u \in \mathbb{R}^n$ and $v \in \mathbb{R}^m$ we set $\dot{u} = (u_{I_{\varepsilon,\kappa}}^\top, u_{I_{\varepsilon,\kappa}^{\complement}}^\top)^\top$ and $\dot{v} = (v_{J_{\varepsilon,\kappa}}^\top, v_{J_{\varepsilon,\kappa}^{\complement}}^\top)^\top$. We then have

$$\Psi_{\varepsilon,\kappa}(u, v) = \mathbf{1}_n^\top \widetilde{B}(\dot{u}, \dot{v})\mathbf{1}_m - \kappa\dot{\mu}^\top \dot{u} - \kappa^{-1}\dot{\nu}^\top \dot{v},$$

and

$$\Psi(u, v) = \mathbf{1}_n^\top \widetilde{B}(\dot{u}, \dot{v})\mathbf{1}_m - \dot{\mu}^\top \dot{u} - \dot{\nu}^\top \dot{v},$$

where

$$\widetilde{B}(\dot{u}, \dot{v}) = \Delta(e^{\dot{u}})\widetilde{K}\Delta(e^{\dot{v}}).$$

Let us consider the convex function

$$(\hat{u}, \hat{v}) \mapsto \langle \mathbf{1}_n, \widetilde{B}(\dot{\hat{u}}, \dot{\hat{v}})\mathbf{1}_m\rangle - \langle \kappa\dot{\hat{u}}, \widetilde{B}(\dot{u}^{\mathrm{sc}}, \dot{v}^{\mathrm{sc}})\mathbf{1}_m\rangle - \langle \kappa^{-1}\dot{\hat{v}}, \widetilde{B}(\dot{u}^{\mathrm{sc}}, \dot{v}^{\mathrm{sc}})^\top\mathbf{1}_n\rangle.$$

Gradient inequality of any convex function g at point $x_o$ reads as $g(x_o) \geq g(x) + \langle \nabla g(x), x_o - x\rangle$, for all $x \in \mathbf{dom}(g)$. Applying the latter fact to the above function at point $(u^\star, v^\star)$ we obtain

$$\langle \mathbf{1}_n, \widetilde{B}(\dot{u}^{\mathrm{sc}}, \dot{v}^{\mathrm{sc}})\mathbf{1}_m\rangle - \langle \kappa\dot{u}^{\mathrm{sc}}, \widetilde{B}(\dot{u}^{\mathrm{sc}}, \dot{v}^{\mathrm{sc}})\mathbf{1}_m\rangle - \langle \kappa^{-1}\dot{v}^{\mathrm{sc}}, \widetilde{B}(\dot{u}^{\mathrm{sc}}, \dot{v}^{\mathrm{sc}})^\top\mathbf{1}_n\rangle$$
$$- \left(\langle \mathbf{1}_n, \widetilde{B}(\dot{u}^\star, \dot{v}^\star)\mathbf{1}_m\rangle - \langle \kappa\dot{u}^\star, \widetilde{B}(\dot{u}^{\mathrm{sc}}, \dot{v}^{\mathrm{sc}})\mathbf{1}_m\rangle - \langle \kappa^{-1}\dot{v}^\star, \widetilde{B}(\dot{u}^{\mathrm{sc}}, \dot{v}^{\mathrm{sc}})^\top\mathbf{1}_n\rangle\right)$$
$$\leq \langle \dot{u}^{\mathrm{sc}} - \dot{u}^\star, (1 - \kappa)\widetilde{B}(\dot{u}^{\mathrm{sc}}, \dot{v}^{\mathrm{sc}})\mathbf{1}_m\rangle + \langle \dot{v}^{\mathrm{sc}} - \dot{v}^\star, (1 - \kappa^{-1})\widetilde{B}(\dot{u}^{\mathrm{sc}}, \dot{v}^{\mathrm{sc}})^\top\mathbf{1}_n\rangle.$$

Moreover,

$$\Psi_{\varepsilon,\kappa}(u^{\mathrm{sc}}, v^{\mathrm{sc}}) - \Psi(u^\star, v^\star) = \langle \mathbf{1}_n, \widetilde{B}(\dot{u}^{\mathrm{sc}}, \dot{v}^{\mathrm{sc}})\mathbf{1}_m\rangle - \langle \kappa\dot{u}^{\mathrm{sc}}, \widetilde{B}(\dot{u}^{\mathrm{sc}}, \dot{v}^{\mathrm{sc}})\mathbf{1}_m\rangle - \langle \kappa^{-1}\dot{v}^{\mathrm{sc}}, \widetilde{B}(\dot{u}^{\mathrm{sc}}, \dot{v}^{\mathrm{sc}})\mathbf{1}_n^\top\rangle$$
$$- \left(\langle \mathbf{1}_n, \widetilde{B}(\dot{u}^\star, \dot{v}^\star)\mathbf{1}_m\rangle - \langle \dot{u}^\star, \widetilde{B}(\dot{u}^{\mathrm{sc}}, \dot{v}^{\mathrm{sc}})\mathbf{1}_m\rangle - \langle \dot{v}^\star, \widetilde{B}(\dot{u}^{\mathrm{sc}}, \dot{v}^{\mathrm{sc}})^\top\mathbf{1}_n\rangle\right)$$
$$+ \langle \kappa\dot{u}^{\mathrm{sc}} - \dot{u}^\star, \widetilde{B}(\dot{u}^{\mathrm{sc}}, \dot{v}^{\mathrm{sc}})\mathbf{1}_m - \dot{\mu}\rangle + \langle \kappa^{-1}\dot{v}^{\mathrm{sc}} - \dot{v}^\star, \widetilde{B}(\dot{u}^{\mathrm{sc}}, \dot{v}^{\mathrm{sc}})^\top\mathbf{1}_n - \dot{\nu}\rangle.$$

Hence,

$$\Psi_{\varepsilon,\kappa}(u^{\mathrm{sc}}, v^{\mathrm{sc}}) - \Psi(u^\star, v^\star) + \left(\langle \mathbf{1}_n, \widetilde{B}(\dot{u}^\star, \dot{v}^\star)\mathbf{1}_m\rangle\right.$$
$$- \langle \dot{u}^\star, \widetilde{B}(\dot{u}^{\mathrm{sc}}, \dot{v}^{\mathrm{sc}})\mathbf{1}_m\rangle - \langle \dot{v}^\star, \widetilde{B}(\dot{u}^{\mathrm{sc}}, \dot{v}^{\mathrm{sc}})^\top\mathbf{1}_n\rangle\right)$$
$$- \langle \kappa\dot{u}^{\mathrm{sc}} - \dot{u}^\star, \widetilde{B}(\dot{u}^{\mathrm{sc}}, \dot{v}^{\mathrm{sc}})\mathbf{1}_m - \dot{\mu}\rangle - \langle \kappa^{-1}\dot{v}^{\mathrm{sc}} - \dot{v}^\star, \widetilde{B}(\dot{u}^{\mathrm{sc}}, \dot{v}^{\mathrm{sc}})^\top\mathbf{1}_n - \dot{\nu}\rangle$$
$$\leq \langle \dot{u}^{\mathrm{sc}} - \dot{u}^\star, (1 - \kappa)\widetilde{B}(\dot{u}^{\mathrm{sc}}, \dot{v}^{\mathrm{sc}})\mathbf{1}_m\rangle + \langle \dot{v}^{\mathrm{sc}} - \dot{v}^\star, (1 - \kappa^{-1})\widetilde{B}(\dot{u}^{\mathrm{sc}}, \dot{v}^{\mathrm{sc}})^\top\mathbf{1}_n\rangle$$
$$+ \left(\langle \mathbf{1}_n, \widetilde{B}(\dot{u}^\star, \dot{v}^\star)\mathbf{1}_m\rangle - \langle \kappa\dot{u}^\star, \widetilde{B}(\dot{u}^{\mathrm{sc}}, \dot{v}^{\mathrm{sc}})\mathbf{1}_m\rangle - \langle \kappa^{-1}\dot{v}^\star, \widetilde{B}(\dot{u}^{\mathrm{sc}}, \dot{v}^{\mathrm{sc}})^\top\mathbf{1}_n\rangle\right).$$

Then,

$$\Psi_{\varepsilon,\kappa}(u^{\mathrm{sc}}, v^{\mathrm{sc}}) - \Psi(u^\star, v^\star) \leq \langle \dot{u}^{\mathrm{sc}} - \dot{u}^\star, (1-\kappa)\widetilde{B}(\dot{u}^{\mathrm{sc}}, \dot{v}^{\mathrm{sc}})\mathbf{1}_m \rangle + \langle \dot{v}^{\mathrm{sc}} - \dot{v}^\star, (1-\kappa^{-1})\widetilde{B}(\dot{u}^{\mathrm{sc}}, \dot{v}^{\mathrm{sc}})^\top \mathbf{1}_n \rangle$$
$$+ \left( \langle \mathbf{1}_n, \widetilde{B}(\dot{u}^\star, \dot{v}^\star)\mathbf{1}_m \rangle - \langle \kappa \dot{u}^\star, \widetilde{B}(\dot{u}^{\mathrm{sc}}, \dot{v}^{\mathrm{sc}})\mathbf{1}_m \rangle - \langle \kappa^{-1}\dot{v}^\star, \widetilde{B}(\dot{u}^{\mathrm{sc}}, \dot{v}^{\mathrm{sc}})^\top \mathbf{1}_n \rangle \right)$$
$$+ \langle \kappa \dot{u}^{\mathrm{sc}} - \dot{u}^\star, \widetilde{B}(\dot{u}^{\mathrm{sc}}, \dot{v}^{\mathrm{sc}})\mathbf{1}_m - \dot{\mu} \rangle + \langle \kappa^{-1}\dot{v}^{\mathrm{sc}} - \dot{v}^\star, \widetilde{B}(\dot{u}^{\mathrm{sc}}, \dot{v}^{\mathrm{sc}})^\top \mathbf{1}_n - \dot{\nu} \rangle$$
$$- \left( \langle \mathbf{1}_n, \widetilde{B}(\dot{u}^\star, \dot{v}^\star)\mathbf{1}_m \rangle - \langle \dot{u}^\star, \widetilde{B}(\dot{u}^{\mathrm{sc}}, \dot{v}^{\mathrm{sc}})\mathbf{1}_m \rangle - \langle \dot{v}^\star, \widetilde{B}(\dot{u}^{\mathrm{sc}}, \dot{v}^{\mathrm{sc}})^\top \mathbf{1}_n \rangle \right),$$

which yields

$$\Psi_{\varepsilon,\kappa}(u^{\mathrm{sc}}, v^{\mathrm{sc}}) - \Psi(u^\star, v^\star) \leq \langle \kappa \dot{u}^{\mathrm{sc}} - \dot{u}^\star, \widetilde{B}(\dot{u}^{\mathrm{sc}}, \dot{v}^{\mathrm{sc}})\mathbf{1}_m - \dot{\mu} \rangle + \langle \kappa^{-1}\dot{v}^{\mathrm{sc}} - \dot{v}^\star, \widetilde{B}(\dot{u}^{\mathrm{sc}}, \dot{v}^{\mathrm{sc}})^\top \mathbf{1}_n - \dot{\nu} \rangle$$
$$+ (1-\kappa)\langle \dot{u}^{\mathrm{sc}}, \widetilde{B}(\dot{u}^{\mathrm{sc}}, \dot{v}^{\mathrm{sc}})\mathbf{1}_m \rangle + (1-\kappa^{-1})\langle \dot{v}^{\mathrm{sc}}, \widetilde{B}(\dot{u}^{\mathrm{sc}}, \dot{v}^{\mathrm{sc}})^\top \mathbf{1}_n \rangle.$$

Applying Holder's inequality gives

$$\Psi_{\varepsilon,\kappa}(u^{\mathrm{sc}}, v^{\mathrm{sc}}) - \Psi(u^\star, v^\star) \leq \|\kappa \dot{u}^{\mathrm{sc}} - \dot{u}^\star\|_\infty \|\widetilde{B}(\dot{u}^{\mathrm{sc}}, \dot{v}^{\mathrm{sc}})\mathbf{1}_m - \dot{\mu}\|_1 + \|\kappa^{-1}\dot{v}^{\mathrm{sc}} - \dot{v}^\star\|_\infty \|\widetilde{B}(\dot{u}^{\mathrm{sc}}, \dot{v}^{\mathrm{sc}})^\top \mathbf{1}_n - \dot{\nu}\|_1$$
$$+ |1-\kappa|\langle \dot{u}^{\mathrm{sc}}, \widetilde{B}(\dot{u}^{\mathrm{sc}}, \dot{v}^{\mathrm{sc}})\mathbf{1}_m \rangle + |1-\kappa^{-1}|\langle \dot{v}^{\mathrm{sc}}, \widetilde{B}(\dot{u}^{\mathrm{sc}}, \dot{v}^{\mathrm{sc}})^\top \mathbf{1}_n \rangle$$
$$\leq \left( \|\dot{u}^{\mathrm{sc}} - \dot{u}^\star\|_\infty + |1-\kappa|\|\dot{u}^{\mathrm{sc}}\|_\infty \right) \|\widetilde{B}(\dot{u}^{\mathrm{sc}}, \dot{v}^{\mathrm{sc}})\mathbf{1}_m - \dot{\mu}\|_1$$
$$+ \left( \|\dot{v}^{\mathrm{sc}} - \dot{v}^\star\|_\infty + |1-\kappa^{-1}|\|\dot{v}^{\mathrm{sc}}\|_\infty \right) \|\widetilde{B}(\dot{u}^{\mathrm{sc}}, \dot{v}^{\mathrm{sc}})^\top \mathbf{1}_n - \dot{\nu}\|_1$$
$$+ |1-\kappa|\langle \dot{u}^{\mathrm{sc}}, \widetilde{B}(\dot{u}^{\mathrm{sc}}, \dot{v}^{\mathrm{sc}})\mathbf{1}_m \rangle + |1-\kappa^{-1}|\langle \dot{v}^{\mathrm{sc}}, \widetilde{B}(\dot{u}^{\mathrm{sc}}, \dot{v}^{\mathrm{sc}})^\top \mathbf{1}_n \rangle$$

where, in the last inequality, we use the facts that $\|\kappa \dot{u}^{\mathrm{sc}} - \dot{u}^\star\|_\infty \leq \|\dot{u}^{\mathrm{sc}} - \dot{u}^\star\|_\infty + |1-\kappa|\|\dot{u}^{\mathrm{sc}}\|_\infty$ and $\|\kappa^{-1}\dot{v}^{\mathrm{sc}} - \dot{v}^\star\|_\infty \leq \|\dot{v}^{\mathrm{sc}} - \dot{v}^\star\|_\infty + |1-\kappa^{-1}|\|\dot{v}^{\mathrm{sc}}\|_\infty$. Moreover, note that

$$\begin{cases} \|\dot{u}^{\mathrm{sc}} - \dot{u}^\star\|_\infty = \|u^{\mathrm{sc}} - u^\star\|_\infty, \\ \|\dot{v}^{\mathrm{sc}} - \dot{v}^\star\|_\infty = \|v^{\mathrm{sc}} - v^\star\|_\infty, \end{cases} \quad \text{and} \quad \begin{cases} \|\widetilde{B}(\dot{u}^{\mathrm{sc}}, \dot{v}^{\mathrm{sc}})\mathbf{1}_m - \dot{\mu}\|_1 = \|B(u^{\mathrm{sc}}, v^{\mathrm{sc}})\mathbf{1}_m - \mu\|_1 = \|\mu^{\mathrm{sc}} - \mu\|_1, \\ \|\widetilde{B}(\dot{u}^{\mathrm{sc}}, \dot{v}^{\mathrm{sc}})^\top \mathbf{1}_n - \dot{\nu}\|_1 = \|B(u^{\mathrm{sc}}, v^{\mathrm{sc}})^\top \mathbf{1}_n - \nu\|_1 = \|\nu^{\mathrm{sc}} - \nu\|_1. \end{cases}$$

Then

$$\Psi_{\varepsilon,\kappa}(u^{\mathrm{sc}}, v^{\mathrm{sc}}) - \Psi(u^\star, v^\star) \leq \left( \|u^{\mathrm{sc}} - u^\star\|_\infty + |1-\kappa|\|u^{\mathrm{sc}}\|_\infty \right) \|\mu^{\mathrm{sc}} - \mu\|_1$$
$$+ \left( \|v^{\mathrm{sc}} - v^\star\|_\infty + |1-\kappa^{-1}|\|v^{\mathrm{sc}}\|_\infty \right) \|\nu^{\mathrm{sc}} - \nu\|_1$$
$$+ |1-\kappa|\langle u^{\mathrm{sc}}, \mu^{\mathrm{sc}} \rangle + |1-\kappa^{-1}|\langle v^{\mathrm{sc}}, \nu^{\mathrm{sc}} \rangle$$
$$\leq \left( \|u^{\mathrm{sc}} - u^\star\|_\infty + |1-\kappa|\|u^{\mathrm{sc}}\|_\infty \right) \|\mu^{\mathrm{sc}} - \mu\|_1$$
$$+ \left( \|v^{\mathrm{sc}} - v^\star\|_\infty + |1-\kappa^{-1}|\|v^{\mathrm{sc}}\|_\infty \right) \|\nu^{\mathrm{sc}} - \nu\|_1 \qquad (4)$$
$$+ |1-\kappa|\|u^{\mathrm{sc}}\|_\infty \|\mu^{\mathrm{sc}}\|_1 + |1-\kappa^{-1}|\|v^{\mathrm{sc}}\|_\infty \|\nu^{\mathrm{sc}}\|_1.$$

Next, we bound the two terms $\|u^{\mathrm{sc}} - u^\star\|_\infty$ and $\|v^{\mathrm{sc}} - v^\star\|_\infty$. If $r \in I_{\varepsilon,\kappa}^{\complement}$, then we have

$$|(u^{\mathrm{sc}})_r - u_r^\star| = \left| \log \left( \frac{\sum_{j=1}^m K_{rj} e^{v_j^\star}}{\sum_{j=1}^m \frac{\kappa \mu_r}{m\varepsilon}} \right) \right|$$
$$\overset{(\star)}{\leq} \left| \log \left( \max_{1 \leq i \leq m} \frac{K_{rj} e^{v_j^\star}}{\frac{\kappa \mu_r}{m\varepsilon}} \right) \right|$$
$$\leq \left| \max_{1 \leq j \leq m} (v_j^\star - \log(\frac{\kappa \mu_r}{m\varepsilon})) \right|$$
$$\leq \|v^\star - \log(\frac{\kappa \mu_r}{m\varepsilon})\|_\infty$$
$$\leq \|v^\star - v^{\mathrm{sc}}\|_\infty + \log(\frac{m\varepsilon^2}{c_{\mu\nu}}).$$

where the inequality $(\star)$ comes from the fact that $\frac{\sum_{j=1}^n a_j}{\sum_{j=1}^n b_j} \leq \max_{1 \leq j \leq n} \frac{a_j}{b_j}, \forall a_j, b_j > 0$. Now, if $r \in I_{\varepsilon,\kappa}$, we get

$$|u_r^{\mathrm{sc}} - u_r^\star| = \left| \log \left( \frac{\kappa \sum_{j=1}^m K_{rj} e^{v_j^\star}}{\sum_{j=1}^m K_{rj} e^{(v^{\mathrm{sc}})_j}} \right) \right| \leq \left| \log \left( \frac{\sum_{j=1}^m K_{rj} e^{v_j^\star}}{\sum_{j=1}^m K_{rj} e^{(v^{\mathrm{sc}})_j}} \right) \right| \overset{(\star)}{\leq} \|v^{\mathrm{sc}} - v^\star\|_\infty.$$

If $s \in J_{\varepsilon,\kappa}^{\complement}$ then

$$
\begin{aligned}
|v_s^{\mathrm{sc}} - v_s^{\star}| &= \left| \log(\varepsilon\kappa) - \log\Big(\frac{\nu_s}{\sum_{i=1}^{n} K_{is} e^{u_i^{\star}}}\Big) \right| \\
&\leq \left| \log \Big( \max_{1 \leq i \leq n} \frac{K_{is} e^{u_i^{\star}}}{\frac{\nu_s}{n\kappa\varepsilon}} \Big) \right| \\
&\overset{(\star)}{\leq} \big| \max_{1 \leq i \leq n} (u_i^{\star} - \log(\frac{\nu_s}{n\kappa\varepsilon})) \big| \\
&\leq \| u^{\star} - \log(\frac{\nu_s}{n\kappa\varepsilon}) \|_{\infty} \\
&\leq \| u^{\star} - u^{\mathrm{sc}} \|_{\infty} + \log(\frac{n\varepsilon^2}{c_{\mu\nu}}).
\end{aligned}
$$

If $s \in J_{\varepsilon,\kappa}$ then

$$
|v_s^{\mathrm{sc}} - v_s^{\star}| = \left| \log \Big( \frac{\kappa \sum_{i=1}^{m} K_{ri} e^{u_i^{\star}}}{\sum_{i=1}^{m} K_{ri} e^{(u^{\mathrm{sc}})_i}} \Big) \right| \leq \left| \log \Big( \frac{\kappa \sum_{i=1}^{m} K_{ri} e^{v_j^{\star}}}{\sum_{i=1}^{m} K_{ri} e^{(u^{\mathrm{sc}})_i}} \Big) \right| \overset{(\star)}{\leq} \| u^{\mathrm{sc}} - u^{\star} \|_{\infty}.
$$

Therefore, we obtain the followoing bound:

$$
\begin{aligned}
\max\{ \| u^{\star} - u^{\mathrm{sc}} \|_{\infty}, \| v^{\star} - v^{\mathrm{sc}} \|_{\infty} \} &\leq \max \Big\{ \| u^{\star} \|_{\infty} + \| u^{\mathrm{sc}} \|_{\infty} + \log(\frac{n\varepsilon^2}{c_{\mu\nu}}), \| v^{\star} \|_{\infty} + \| v^{\mathrm{sc}} \|_{\infty} + \log(\frac{m\varepsilon^2}{c_{\mu\nu}}) \Big\} \\
&\leq 2 \Big( \| u^{\star} \|_{\infty} + \| v^{\star} \|_{\infty} + \| u^{\mathrm{sc}} \|_{\infty} + \| v^{\mathrm{sc}} \|_{\infty} + \log \big( \frac{(n \vee m)\varepsilon^2}{c_{\mu\nu}} \big) \Big).
\end{aligned}
\tag{5}
$$

Now, Lemma 3.2 in [4] provides an upper bound for the $\ell_{\infty}$ of the optimal solution pair $(u^{\star}, v^{\star})$ of problem (2) as follows: $\| u^{\star} \|_{\infty} \leq A$ and $\| v^{\star} \|_{\infty} \leq A$, where

$$
A = \frac{\| C \|_{\infty}}{\eta} + \log \big( \frac{n \vee m}{c_{\mu\nu}^2} \big).
\tag{6}
$$

Plugging (5) and (6) in (4), we obtain

$$
\begin{aligned}
\Psi_{\varepsilon,\kappa}(u^{\mathrm{sc}}, v^{\mathrm{sc}}) - \Psi(u^{\star}, v^{\star}) \leq{}& 2 \big( A + \| u^{\mathrm{sc}} \|_{\infty} + \| v^{\mathrm{sc}} \|_{\infty} + \log \big( \frac{(n \vee m)\varepsilon^2}{c_{\mu\nu}} \big) \big) \big( \| \mu^{\mathrm{sc}} - \mu \|_1 + \| \nu^{\mathrm{sc}} - \nu \|_1 \big) \\
&+ |1 - \kappa| \big( \| u^{\mathrm{sc}} \|_{\infty} \| \mu^{\mathrm{sc}} \|_1 + \| \mu^{\mathrm{sc}} - \mu \|_1 \big) \\
&+ |1 - \kappa^{-1}| \big( \| v^{\mathrm{sc}} \|_{\infty} \| \nu^{\mathrm{sc}} \|_1 + \| \nu^{\mathrm{sc}} - \nu \|_1 \big).
\end{aligned}
\tag{7}
$$

By Proposition 1, we have

$$
\| u^{\mathrm{sc}} \|_{\infty} \leq \log \big( \frac{\varepsilon}{\kappa} \vee \frac{1}{m\varepsilon K_{\min}} \big) \text{ and } \| v^{\mathrm{sc}} \|_{\infty} \leq \log \big( \varepsilon\kappa \vee \frac{1}{n\varepsilon K_{\min}} \big)
$$

and hence by Remark 1,

$$
\| u^{\mathrm{sc}} \|_{\infty} = \mathcal{O} \big( \log(n^{1/4}/(mK_{\min})^{3/4} c_{\mu\nu}^{1/4}) \big) \text{ and } \| u^{\mathrm{sc}} \|_{\infty} = \mathcal{O} \big( \log(m^{1/4}/(nK_{\min})^{3/4} c_{\mu\nu}^{1/4}) \big).
$$

Acknowledging that $\log(1/K_{\min}^2) = 2\| C \|_{\infty}/\eta$, we have

$$
A + \| u^{\mathrm{sc}} \|_{\infty} + \| v^{\mathrm{sc}} \|_{\infty} + \log \big( \frac{(n \vee m)\varepsilon^2}{c_{\mu\nu}} \big) \big) = \mathcal{O} \Big( \frac{\| C \|_{\infty}}{\eta} + \log \big( \frac{(n \vee m)^2}{nm c_{\mu\nu}^{7/2}} \big) \Big).
$$

Letting $\Omega_{\kappa} := |1 - \kappa| \big( \| u^{\mathrm{sc}} \|_{\infty} \| \mu^{\mathrm{sc}} \|_1 + \| \mu^{\mathrm{sc}} - \mu \|_1 \big) + |1 - \kappa^{-1}| \big( \| v^{\mathrm{sc}} \|_{\infty} \| \nu^{\mathrm{sc}} \|_1 + \| \nu^{\mathrm{sc}} - \nu \|_1 \big)$. We have that

$$
\begin{aligned}
\Omega_{\kappa} &= \mathcal{O} \Big( \big( \frac{\| C \|_{\infty}}{\eta} + \log \big( \frac{1}{(nm)^{3/4} c_{\mu\nu}^{1/2}} \big) \big) \big( |1 - \kappa| (\| \mu^{\mathrm{sc}} \|_1 + \| \mu^{\mathrm{sc}} - \mu \|_1) + |1 - \kappa^{-1}| (\| \nu^{\mathrm{sc}} \|_1 + \| \nu^{\mathrm{sc}} - \nu \|_1) \big) \Big) \\
&= \mathcal{O} \Big( \big( \frac{\| C \|_{\infty}}{\eta} + \log \big( \frac{(n \vee m)^2}{nm c_{\mu\nu}^{7/2}} \big) \big) \big( |1 - \kappa| \| \mu^{\mathrm{sc}} \|_1 + |1 - \kappa^{-1}| \| \nu^{\mathrm{sc}} \|_1 + |1 - \kappa| + |1 - \kappa^{-1}| \big) \Big).
\end{aligned}
$$

Hence, we arrive at

$$\Psi_{\varepsilon,\kappa}(u^{\mathrm{sc}}, v^{\mathrm{sc}}) - \Psi(u^{\star}, v^{\star}) = \mathcal{O}\big(R(\|\mu - \mu^{\mathrm{sc}}\|_1 + \|\nu - \nu^{\mathrm{sc}}\|_1 + \omega_\kappa)\big).$$

$\square$

To more characterize $\omega_\kappa$, the following lemma expresses an upper bound with respect to $\ell_1$-norm of $\mu^{\mathrm{sc}}$ and $\nu^{\mathrm{sc}}$.

**Lemma 2.** *Let* $(u^{sc}, v^{sc})$ *be an optimal solution of problem* (6). *Then one has*

$$\|\mu^{sc}\|_1 = \mathcal{O}\Big(\frac{n_b\sqrt{m}}{\sqrt{n}K_{\min}c_{\mu\nu}} + (n - n_b)\Big(\frac{m_b}{\sqrt{nm}c_{\mu\nu}K_{\min}^{3/2}} + \frac{m - m_b}{\sqrt{nm}K_{\min}}\Big)\Big), \qquad (8)$$

*and*

$$\|\nu^{sc}\|_1 = \mathcal{O}\Big(\frac{m_b\sqrt{n}}{\sqrt{m}K_{\min}c_{\mu\nu}} + (m - m_b)\Big(\frac{n_b}{\sqrt{nm}c_{\mu\nu}K_{\min}^{3/2}} + \frac{n - n_b}{\sqrt{nm}K_{\min}}\Big)\Big). \qquad (9)$$

*Proof.* Using inequality (8), we obtain

$$\|\mu^{\mathrm{sc}}\|_1 = \sum_{i \in I_{\varepsilon,\kappa}} \mu_i^{\mathrm{sc}} + \sum_{i \in I_{\varepsilon,\kappa}^{\complement}} \mu_i^{\mathrm{sc}}$$

$$\overset{(2)}{=} \kappa\|\mu_{I_{\varepsilon,\kappa}}^{\mathrm{sc}}\|_1 + \frac{\varepsilon}{\kappa} \sum_{i \in I^{\complement}} \Big( \sum_{j \in J_{\varepsilon,\kappa}} K_{ij}e^{v_j^{\mathrm{sc}}} + \varepsilon\kappa \sum_{j \in J_{\varepsilon,\kappa}^{\complement}} K_{ij} \Big)$$

$$\overset{(8)}{\le} \kappa\|\mu_{I_{\varepsilon,\kappa}}^{\mathrm{sc}}\|_1 + (n - n_b)\Big(\frac{m_b \max_{j \in J_{\varepsilon,\kappa}} \nu_j}{n\kappa K_{\min}} + (m - m_b)\varepsilon^2\Big).$$

Using Remark 1, we get the desired closed form in (8). Similarly, we can prove the same statement for $\|\nu^{\mathrm{sc}}\|_1$. $\square$

# 5   Additional experimental results

**Experimental setup.**   All computations have been run on each single core of an Intel Xeon E5-2630 processor clocked at 2.4 GHz in a Linux machine with 144 Gb of memory.

**On the use of a constrained L-BFGS-B solver.**   It is worth to note that standard Sinkhorn's alternating minimization cannot be applied for the constrained screened dual problem (6). This appears more clearly while writing its optimality conditions (see Equations (2) and (3) ). We resort to a L-BFGS-B algorithm to solve the constrained convex optimization problem on the screened variables (6), but any other efficient solver (e.g., proximal based method or Newton method) could be used. The choice of the starting point for the L-BFGS-B algorithm is given by the solution of the RESTRICTED SINKHORN method (see Algorithm 1), which is a Sinkhorn-like algorithm applied to the active dual variables. While simple and efficient the solution of this RESTRICTED SINKHORN algorithm does not satisfy the lower bound constraints of Problem (6). We further note that, as for the SINKHORN algorithm, our SCREENKHORN algorithm can be accelerated using a GPU implementation[1] of the L-BFGS-B algorithm [3].

**Comparison with other solvers.**   We have considered experiments with GREENKHORN algorithm [2] but the implementation in POT library and our Python version of Matlab Altschuler's Greenkhorn code[2] were not competitive with SINKHORN. Hence, for both versions, SCREENKHORN is more competitive than GREENKHORN. The computation time gain reaches an order of 30 when comparing our method with GREENKHORN while SCREENKHORN is almost 2 times faster than SINKHORN.

**Algorithm 1:** RESTRICTED SINKHORN

1. **set:** $\bar{f}_u = \varepsilon\kappa\, c(K_{I_{\varepsilon,\kappa},J_{\varepsilon,\kappa}^{\complement}}), \bar{f}_v = \varepsilon\kappa^{-1}\, r(K_{I_{\varepsilon,\kappa}^{\complement},J_{\varepsilon,\kappa}})$;
2. **for** $t = 1, 2, 3$ **do**
   $f_v^{(t)} \leftarrow K_{I_{\varepsilon,\kappa},J_{\varepsilon,\kappa}}^{\top}\, u + \bar{f}_v$;
   $v^{(t)} \leftarrow \frac{\nu_{J_{\varepsilon,\kappa}}}{\kappa f_v^{(t)}}$;
   $f_u^{(t)} \leftarrow K_{I_{\varepsilon,\kappa},J_{\varepsilon,\kappa}}\, v + \bar{f}_u$;
   $u^{(t)} \leftarrow \frac{\kappa\mu_{I_{\varepsilon,\kappa}}}{f_u^{(t)}}$;
   $u \leftarrow u^{(t)}, v \leftarrow v^{(t)}$;
3. **return** $(u^{(t)}, v^{(t)})$

Figure 1: $\frac{T_{\text{GREENKHORN}}}{T_{\text{SCREENKHORN}}}$: Running time gain for the toy problem (see Section 5.2) as a function of the data decimation factor in SCREENKHORN, for different settings of the regularization parameter $\eta$.

Figure 2: Accuracy of a 1-nearest-neighbour after WDA for the (left) toy problem and, (right) MNIST). We note a slight loss of performance for the toy problem, whereas for MNIST, all approaches yield the same performance.

Figure 3: (top-left) Accuracy and (bottom-right) computational time gain on the toy dataset for $\eta = 0.1$ and 1-nearest-neighbour. (bottom) accuracy and gain but for a 5-nearest-neighbour. We can note that a slight loss of performances occur for larger training set sizes especially for 1-nearest-neighbour. Computational gains increase with the dataset size and are on average of the order of magnitude.

Figure 4: OT Domain Adaptation on a 3-class Gaussian toy problem. (top-left) Examples of source and target samples. (top-right) Evolution of the accuracy of a 1-nearest-neighbour classifier with respect to the number of samples. (bottom-left) Running time of the SINKHORN and SCREENKHORN for different decimation factors. (bottom-right). Gain in computation time. This toy problem is a problem in which classes are overlapping and distance between samples are rather limited. According to our analysis, this may be a situation in which SCREENKHORN may result in smaller computational gain. We can remark that with respect to the accuracy SCREENKHORN with decimation factors up to 10 are competitive with SINKHORN, although a slight loss of performance. Regarding computational time, for this example, small decimation factors does not result in gain. However for above 5-factor decimation, the gain goes from 2 to 10 depending on the number of samples.

Figure 5: OT Domain adaptation MNIST to USPS : (top) Accuracy and (bottom) running time of SINKHORN and SCREENKHORN for hyperparameter of the $\ell_{p,1}$ regularizer (left) $\lambda = 1$ and (right) $\lambda = 10$. Note that this value impacts the ground cost of each Sinkhorn problem involved in the iterative algorithm. The accuracy panels also report the performance of a 1-NN when no-adaptation is performed. We remark that the strenght of the class-based regularization has influence on the performance of SCREENKHORN given a decimation factor. For small value on the left, SCREENKHORN slightly performs better than SINKHORN, while for large value, some decimation factors leads to loss of performances. Regarding, running time, we can note that SINKHORN is far less efficient than SCREENKHORN with an order of magnitude for intermediate number of samples.

Figure 6: OT Domain adaptation MNIST to USPS : (left) Accuracy and (right) running time of SINKHORN and SCREENKHORN for the best performing (on average of 10 trials) hyperparameter $\ell_{p,1}$ chosen among the set $\{0.1, 1, 5, 10\}$. We can note that in this situation, there is not loss of accuracy while our SCREENKHORN is still about an order of magnitude more efficient than Sinkhorn.

## Footnotes

[1] https://github.com/nepluno/lbfgsb-gpu

[2] https://github.com/JasonAltschuler/OptimalTransportNIPS17