[Reviews · NeurIPS 2019]

Reviewer 1



This paper proposes a reformulation of the dual of the entropy regularized wassserstein distance problem that is amenable to screening techniques. Such techniques allow to reduce the dimension of optimization problems hence reducing the computational costs. Here static screening rule is proposed, meaning that the variables are screened before running the solver which is here an L-BFGS-B quasi-Newton method. Two screening techniques are proposed, either using a fixed threshold or a fixed budget. The latter appearing easier to use. A theorem quantifying the error on the orignal problem induced by approximation and screening is provided. The new solver is then employed in 2 ML problems (WDA and OTDA). I have no strong concerns about the theoretical derivations that recall the basics of regularized transport and progressively introduces the the contributions. I am more concerned about the experiments - The method is not compared to Greenkhorn or any improved solver for this optimization problems. - The beauty of Sinkhorn is that it can be accelerated by GPU computations. It is unclear how the new algorithm would compete with a GPU-accelerated Sinkhorn algorithm. - The term "somewhat equivalent" at the end of the paper is not very scientific. - In Fig 6 even the smallest decimation leads to an immediate drop of performance. This suggests a fundamental and uncontroled difference in the solvers used. This observation is consistent with Figure 7 (low left on OTDA toy) where a decimation of 2 or less is slower to converge than a regular Sinkhorn. - There are quite weird results in figure 8 too where a decimation of 1.5 or 2 leads to a drop in accuracy while the more strigent decimation factors do not incure any performance drop. Minor - For figures when the legend and lines follow an order you should use a continuous paletter of colors and not categorical colors like here. It's then easier to see the gradient of speed gain with decimation. Typos - l19 : improve -> improved - quasi Newton -> quasi-Newton - l104: referred as Sinkhorn algorithm -> referred to as Sinkhorn algorithm

Reviewer 2



Edit post feedback: I thank the authors for their thorough response to my questions and concerns. I have raised my rating from the current placeholder to a full accept. === Summary. Due to many recent applications, it is an important problem to develop scalable algorithms for approximating OT and Sinkhorn Divergences. This paper approaches this problem by proposing a pre-processing step in which they hope to identify dual variables with small optimal value. They then set such variables to a threshold value, and compute the Sinkhorn divergence by only optimizing over the remaining variables. The point is that if the proposed screening step does not affect the value of the Sinkhorn divergence, then this approach should be much more scalable. They then solve the resulting optimization via constrained L-BFGS (as opposed to standard methods such as Sinkhorn's alternating minimization). However, I have several concerns, the primary being I am not convinced this screening pre-processing step yields a quantity that is actually close to the desired Sinkhorn divergence/OT. The main theoretical guarantee of the paper is Proposition 3, which bounds this deviation. There are several issues with this bound. I first note that bound is not easily interpretable—it is significantly preferable (and should be changed as such) for such a bound to be re-phrased as something like: “the deviation is at most delta, if the screening parameters (eps and kappa) are set to ___ values. Then for these values, the algorithms runs in time ___ .” Nevertheless, let me raise my concerns while trying to interpret this Proposition 3 bound as written: =1. The first quantity in the bound, C_max/eta, is only small when eta is large, which is the highly regularized regime. But in this regime, the Sinkhorn divergence is a poor estimate of OT. Indeed, it is known that eta must be taken of size Theta(delta/log n) for the Sinkhorn divergence to be within +- delta of the OT value. But in this regime, the C_max/eta term in your bound scales in the accuracy delta extremely poorly as 1/delta. For example, in practice, eta is typically taken to be around 1/10 or 1/20 (if not smaller). Even in this so-called “moderately regularized regime”, C_max/eta is extremely large, e.g. 10*C_max or 20*C_max… (Of course there is the hidden constant in the big-O in your bound. It would be helpful to know what this constant is, but I suspect it is not much smaller than 1, if at all. This would be unfortunate since approximating OT within C_max is trivial, and can be done without even looking at the data: simply output the independent coupling mu*nu^T.) To summarize, in the regularization regime where the Sinkhorn divergence actually approximates OT well (i.e. delta << 1 small, and eta~=del), the error in the presented bound appears to be so big (namely C_max/del), to the extent that it is perhaps bigger than C_max (which would then be a meaningless bound, see above comment). Am I missing something? =2. I am concerned that the terms ||mu-mu^sc||_1 and ||nu-nu^sc||_1 in the bound, can be large. My concern arises from the fact that for matrix scaling (which is exactly the dual Sinkhorn divergence), there are matrices for which the dual optimal solutions u and v are s.t. the range of e^{u_i} and e^{v_j} is exponentially large in n. See e.g., page 10 of the paper Kalantari-Khachiyan 1996 “On the complexity of non-negative matrix scaling” (For further discussion of such issues, see also e.g. Cohen et al. 2017 “Matrix scaling and balancing…”) From what I remember, these hard examples also apply for `approximate’ scaling, i.e. whenever u,v do not have exponentially large ranges, then diag(e^u) K diag(e^v) has marginals which are very far from the desired marginals mu,nu. In other words, for such inputs, ||mu-mu^sc||_1 and ||nu-nu^sc||_1 can only be small if the thresholds in (4) are exponentially small in n. But this requires taking your screening parameter eps to be exponentially small in n, which means that c_{mu,nu} will also be exponentially small in n, yielding a dependence of \log(1/c_{mu,nu}) = poly(n) in the Prop 3 bound, which is an extremely massive error bound. Am I missing something? =3. It is unsatisfactory that “\omega_kappa” in the bound is not defined in the proposition statement, and is only stated to decay as o(1) as the screening parameter \kappa tends to 1. It is critical to understand how fast this error term decays in terms of this parameter choice, since the point of this proposed screening approach is to trade off runtime (i.e. screen more aggressively, partially done by taking kappa *far* from 1) with accuracy (i.e. ensure screening does not change the value of the Sinkhorn divergence much, partially done by taking kappa *close* to 1). == Other comments — The proposed approach is to solve the screened (i.e. smaller-size) Sinkhorn divergence problem via L-BGFS. Why not use the standard Sinkhorn alternating minimization typically used for Sinkhorn divergences? Changing two things (screening for pre-processing, and L-BFGS for solving) makes it difficult for the reader to understand the affect of each change. — Something seems a bit suspect about the proposed formulation (3), in that (unless you set the parameter kappa=1, which you purposely do not do) this optimization problem is not invariant under adding a constant times the all-ones vector to u, and subtracting the same from v. This is an important feature of matrix scaling / standard Sinkhorn divergences. — Section 3: the plots are not reproducible (or as informative as they should be) since it is not stated what the distribution of x_i and y_j are/look like. — L45: I suggest changing “reformulation” to a “new formulation”, as the proposed formulation is not the same as Sinkhorn divergences, but rather is an altered formulation of it. (This is written a few times in the paper; the same comment applies throughout.) -- I believe the term log(1/K_min^2) in the bound can be removed, as it is equal to log(1/e^{-2*C_max/eta}) = 2 C_max / eta, and thus absorbed by the first term in the bound. — L64: that is entropy, not negative entropy — When describing the preliminaries in Section 2, it should be briefly mentioned that everything there is expository. — End of first paragraph: technically, the LP solver in [Lee-Sidford 2014] can solve OT exactly in \tilde{O}(n^(2.5)) time. Granted, there is currently no practical implementation of this complicated algorithm, but it should be mentioned there while you are describing runtimes of existing methods. === Comments on exposition: Throughout, there are a great number of typos and grammatical errors. The paper would benefit significantly from careful proofreading. A few examples are: — Abstract, first sentence: “approximate” —> “approximating” — Abstract, second sentence: “neglectible” is misspelled — Abstract: last sentence: “or” —> “and” — L31-32: sentence should be re-written — L33: "using Nystrom method" --> "using the Nystrom method", and missing accent — L34: “considered to approximate” —> “considered for approximating” — L60: “Section 4 devotes to” —> “Section 4 is devoted to” — L103: “referred [to] as” — L103-104: problems with tense — L113: “let’s us” —> “let us” — Fig 1 caption: “2” —> “(2)” — L129: “latter” —> “later” — L422: “rearrangement” is misspelled -- References: missing capitalization in some (but not all) titles, e.g. pca, wasserstein, lagrangian, etc. — etc.

Reviewer 3



Added comment post feedback: The authors propose to add a comparison with Greenkhorn which addressed my main criticism; hoping that the comparison will not be overly obfuscated by differences in implementation. The main concerns of reviewer 2 appear to be very relevant and I am curious to see whether reviewer 2 is satisfied with the authors' feedback. I maintain my rating (i.e. I am in favor of acceptance for this paper but wouldn't be upset if it is rejected). The proposed method is novel, it can be interesting to approximate the Sinkhorn divergence even in the case where eta is not that small (i.e. even if the Sinkhorn divergence itself is a poor approximation for the Kantorovich-Wasserstein distance), and I find the paper honest about the fact that the computational gains might come at the cost of approximation error that strongly depends on eta (i.e. Figure 3). The proposed theoretical results might have flaws (i.e. unknown constants in the bounds) but it can be the starting point for further work. === The regularized OT problem can be solved with Sinkhorn iterations with a computational cost in n times m, where n and m are the numbers of atoms in the input measures. By considering the dual problem, and by screening/thresholding some of the dual variables before performing the optimization, the authors reduce the cost to n_b times m_b, where n_b and m_b are chosen by the user. This comes with an additional approximation error, which is analyzed theoretically. Experiments are provided in toy settings and in a realistic ML pipeline. The article is interesting and participates to a growing literature on fast approximations of optimal transport. The experiments are quite convincing: the proposed method can provide a way of reducing the cost while maintaining a reasonable accuracy. The gains are not always groundbreaking but clearly significant and the method appears to be widely applicable. It is also very nice that users can specify the approximation parameters in terms of n_b and m_b, rather than epsilon and kappa. One criticism is that the method is compared only with the "vanilla" Sinkhorn algorithm, so it is not easy to see whether the proposed method would provide gains compared to other recently proposed variants, such as the Greenkhorn algorithm mentioned in the introduction. It is also unclear whether the proposed thresholding strategy could be combined with these alternative techniques, e.g. with Greenkhorn, for further gains, or whether it is incompatible.

[Author Response · NeurIPS 2019]

**Loss of performances in Figures 6, 8 (Reviewer #1).**    After investigation, those losses are due to hyperparameter choices for the non-convex WDA and OTDA problems. When appropriately selected for each model (decimation factor), we obtain running time gain of same orders without compromising performances. In the final version, we will add to the supplementary new figures related to the regularization path computations and resulting accuracies.

**Comparison with other solvers (Reviewers #1 and #3).**    We have considered experiments with Greenkhorn algorithm but the implementation in POT library and our custom Python version of Matlab Altschuler's Greenkhorn code were not competitive with Sinkhorn. Hence, for both versions, Screenkhorn is more competitive than Greenkhorn. The computation time gain reaches an order of 30 when comparing our method with Greenkhorn while Screenkhorn is almost 2 times faster than Sinkhorn,We will provide this comparison and discussion in the final version.

**On the use of constrained L-BFGS (Reviewers #2 and #3).**    Our proposed screened dual problem given in (3) or (6) involves explicit box constraints on $e^{u_i^{sc}}$ and $e^{v_i^{sc}}$ (see Proposition 1). Hence, it is a constrained smooth optimization problem, and standard Sinkhorn's alternating minimization can not be applied. This appears more clearly while writing its optimality conditions. We resort to L-BFGS-B to solve our constrained convex optimization problem, but any efficient solver (e.g. proximal based method or Newton method) can be used. Notice that as for the Sinkhorn algorithm, our Screenkhorn can be accelerated using a GPU implementation of L-BFGS-B [2].

**Main concerns of Reviewer #2.**    *Concern 1.* The bound in Proposition 3 is similar, up to the additive term $\omega_\kappa$ (a discussion about $\omega_\kappa$ is provided in below), to the ones found in the literature; in particular for the Sinkhorn algorithm (see Lemma 2 in [1]) and for the Greenkhorn algorithm (see Corollary 3.3 in [4]). More formally, letting $\{(u^k, v^k)\}_{k \geq 1}$ denote the iterates returned by the Sinkhorn or the Greenkhorn algorithm, they have $\Psi(u^k, v^k) - \Psi(u^\star, v^\star) = \mathcal{O}(RE^{\overline{k}})$ where $E^k = ||B(u^k, v^k)\mathbf{1} - \mu||_1 + ||B(u^k, v^k)^\top \mathbf{1} - \nu||_1$, and $R = C_{max}/\eta + \log(n) - 2\log(c_{\mu\nu})$ which comes from an upper bound for the $\ell_\infty$-norm of the optimal pair solution $(u^\star, v^\star)$ of Sinkhorn divergence. In our case, supposing that $n = m$ and acknowledging that $\log(1/K_{min}^2) = 2C_{max}/\eta$, we have $R = C_{max}/\eta - 3.5\log(c_{\mu\nu})$. Additionally, we give in Proposition 2 a bound on $E^k$ that becomes small as the sample budget increases.

*Concern 2.* The new formulation (3) has the form of $(\kappa\mu, \nu/\kappa)$-scaling problem under constraints on the variables $u$ and $v$ and the problem is not invariant anymore. This differs significantly from the standard scaling-problems [3], though the sought transportation map $P$ takes a matrix-scaling form. We further emphasize that $\kappa$ plays a key role (that we will emphasize in the final version) in our screening strategy for the dual of Sinkhorn divergence. Indeed, without $\kappa$, $e^u$ and $e^v$ can have inversely related scale that may lead in, for instance $e^u$ being too large and $e^v$ being too small, situation in which the screening test would apply only to coefficients of $e^u$ or $e^v$ and not for both of them. In addition note that given $n$, the bounds in Propositions 2 and 3 are derived using the following control of the parameter $\varepsilon$, which is induced by the screening test's construction (4), $c_{\mu\nu}^{1/4}/\sqrt{n} \leq \varepsilon \leq 1/\sqrt{nK_{\min}}$.

*Concern 3.* An explicit form of $\omega_\kappa$ (with $\omega_1 = 0$) is given in L449 of the paper. In the setting of $n = m$ and using the upper bounds of $||u^{sc}||_\infty$ and $||v^{sc}||_\infty$ in L447, we derive the following bound: $\omega_k \lesssim R'((1 - \kappa)||\mu^{sc}||_1 + (1 - \kappa^{-1})||\nu^{sc}||_1)$ where $R' = C_{\max}/\eta - 0.5\log(n) - 0.5\log(c_{\mu\nu})$. A control for the $\ell_1$-norms of the screened marginals $\mu^{sc}$ and $\nu^{sc}$ are given in Equations (18) and (19) in Lemma 3. Using the bound of the term $\omega_\kappa$, we will clarify the bound in Proposition 3 for the final version of the paper.

**Minor comments (all Reviewers).**    The final version of the paper will include all suggested modifications.

# References

[1] P. Dvurechensky, A. Gasnikov, and A. Kroshnin. Computational optimal transport: Complexity by accelerated gradient descent is better than by Sinkhorn's algorithm. volume 80 of *PMLR*, pages 1367–1376, 2018.

[2] Y. Fei, G. Rong, B. Wang, and W. Wang. Parallel L-BFGS-B algorithm on GPU. *Computers & Graphics*, 40:1 – 9, 2014.

[3] B. Kalantari and L.Khachiyan. On the complexity of nonnegative-matrix scaling. *Linear Algebra and its Applications*, 240:87 – 103, 1996.

[4] T. Lin, N. Ho, and M. Jordan. On efficient optimal transport: An analysis of greedy and accelerated mirror descent algorithms. volume 97 of *PMLR*, pages 3982–3991, 2019.


[Meta-Review · NeurIPS 2019]

This paper proposes a reformulation of the dual of the entropy-regularized optimal transport problem that makes it amenable to screening techniques. These techniques may lead to reduced dimension of the problem, and thus reduced computational costs. Two screening techniques are proposed, either using a fixed threshold or a fixed budget, and a theoretical error analysis is presented. The post-rebuttal consensus view among the reviewers is positive, and this paper is a good addition to the growing literature on computational optimal transport.